# Atmospheric Patterns in Porto Velho, Rondônia, Southwestern Amazon, in a Rhythmic Context between 2017 and 2018

**Graziela T. Tejas** [1] , **Dorisvalder D. Nunes** [2], **Reginaldo M. S. Souza** [1], **Carlos A. S. Querino** [3] , **Marlon R. Faria** [4] ,
**Daiana C. B. Floresta** [2], **Emerson Galvani** [5] , **Michel Watanabe** [2] **and João P. A. Gobo** [2,*]

[1] Civil Engineering Department, Federal Institute of Education, Science and Technology of Rondonia,
Porto Velho 76820-441, Brazil; graziela.tejas@ifro.edu.br (G.T.T.); reginaldo.martins@ifro.edu.br (R.M.S.S.)

[2] Department of Geography, Core of Exact Earth Sciences, Federal University of Rondônia,
Porto Velho 76801-059, Brazil; dorisval@unir.br (D.D.N.); daiana.cristina513@live.com (D.C.B.F.);
michelwatanabe@unir.br (M.W.)

[3] Institute of Education, Agriculture and Environment, Federal University of Amazonas,
Humaitá 69800-000, Brazil; carlosquerino@ufam.edu.br

[4] Institute of Physics, State University of São Paulo, Sao Paulo 05508-090, Brazil; marlon.faria@usp.br

[5] Department of Geography, State University of São Paulo, Sao Paulo 05508-090, Brazil; egalvani@usp.br

\* Correspondence: joao.gobo@unir.br

**Abstract:** This paper aims to analyze the weather conditions in Porto Velho (Rondonia, Brazil, Western Amazon) and the influence of air masses on the climatic elements between 2017 and 2018, using rhythmic analysis. Climatic data were obtained through the official weather station, tabulated and statistically organized, and processed in R Studio programming language. The monitoring of air masses occurred through the synoptic charts of the Navy Hydrography Center. The results were analyzed by dry–rainy transition season, rainy season, wet–dry transition season, and dry season. Thus, the results point out that the Tropical Continental mass (mTc) acted up to 62.9%, responsible for the low precipitation index in October 2017. Although the mass has characteristics of warm and unstable weather, it is even lower than the action of the mEc. In January 2018, there was an 85.5% prevalence of the Continental Equatorial Mass (mEc), added to the action of the South Atlantic Convergence Zone (ZCAS), which contributed to an accumulated rainfall of 443 mm/month. In April 2018, the mEC acted with 56.7%, reaching 35.5% in August. Another highlight was the performance of the Tropical Atlantic mass (mTa) (27.4%) and mTc (19.4%), both of which had a crucial role in the dry season, followed by the Polar Atlantic mass (mPa) (17.7%), that contributed to the phenomenon of "coldness" in the region. Therefore, the mEc is extremely important in the control of the relative humidity of the air and the precipitations, while the mTc is a dissipator of winds that, at times, inhibits the performance of the mEc.

**Keywords:** air masses; rhythmic analysis; geographic climatology; Porto Velho (RO)





## 1. Introduction

The Legal Amazon corresponds to 60% of the Brazilian territory, still covered mainly by forest and diverse plant formations. It plays a fundamental role in the Brazilian and regional South American atmospheric circulation [1–3]. The atmospheric activity of the Amazon gives a substantial variation in seasonal atmospheric weather, especially in temperature and rainfall. This variation is due to the performance of the equatorial system and the atmospheric changes and reorganizations of tropical and extratropical systems when they arrive in the region—it also occurs in Porto Velho [4–6].

Porto Velho is located in the southwest portion of the Western Amazon. Capital of the state of Rondonia, its territorial formation is related to migratory outbreaks arising from local economic cycles to meet external demands. These outbreaks are the rubber cycles, the Colonization Projects derived from the National Integration Program, mining

on the Madeira River, and revisionist ecological programs of the degradation caused by the 1970s occupation. At the beginning of the 21st century, a new economic-migration cycle began: the energy cycle. This cycle is related to the construction of the Santo Antonio and Jirau hydroelectric plants, installed on the Madeira River, which aimed to guarantee the production of energy to Brazil, both developmental operations of the Federal Government's Growth Acceleration Program, known as PAC [7–9]. The reflexes of these economic cycles were an urban demographic explosion, which in the decades of 1990, 2000, and 2010 had an increase in the order of 186%, 38%, and 43% [10,11], and vigorous growth of its urban spot, which, between 1985 and 2012, showed an increase of 145, 72%, reaching 11,158.53 ha in the last year of this interstice [12]. Thus, the city has undergone an intense artificialization of the natural landscape due to its economic cycles and, possibly, has changed atmospheric weather and local climate [13].

The investigation of the habitual behavior of the air masses and its influence on the climate is the object of the studies whose methodological theoretical basis centers on rhythmic analysis [14–17]. Rhythm is the essence of Brazilian geographical climatology, which is the interaction between atmospheric circulation and the elements of the climate, analyzed on a daily chronological basis [18,19]. These are investigations established through the climate rhythm proposed by Monteiro [20]. It involves the daily monitoring of atmospheric systems with data on the volume of precipitation, atmospheric pressure, relative humidity, and temperatures (maximum, average, and minimum), with the possibility of presenting a standard year or an exception, which would be irregularities. It is, therefore, a geographical study developed and refined by the Brazilian school of climatology. It is an approach which began around 1960 that aims to understand the climate through the relationship between meteorological variables and the dynamics of the atmosphere. It is, therefore, a method whose objective is to quantitatively and qualitatively understand the genesis of the action of atmospheric systems through climatic elements at the level of the troposphere, an anthropogenic layer in interaction with the organization of space and the daily life of societies [21]. In order to understand the climatic rhythm in a given location or region, data are collected on atmospheric pressure, temperature, humidity, precipitation, and winds from one or more weather stations. These data are then organized into graphs that show the intensity of each weather element monitored on a daily basis during the period of time that is to be analyzed. Subsequently, the atmospheric systems prevalent on each of the monitoring days are identified. Finally, these data are compared in order to understand the dynamics of the surface atmospheric circulation in terms of its impacts and/or interactions with the socio-environmental aspects being researched.

Therefore, this study analyzed the meteorological conditions in Porto Velho, Rondônia, the largest city in the Western Amazon, through the temporal spatialization of air masses and the rhythmic analysis of climatic elements during one year, from October 2017 to August 2018, following the methodology defined by Monteiro [20].

## 2. Materials and Methods

### 2.1. Study Area

Porto Velho, Rondônia, has an estimated area of 117.34 km$^2$, localized in Southwestern Amazon. The city is subdivided into four zones: the Central Zone, a nucleus of initial occupation that extends from the banks of the Madeira River to the BR-364; the North Zone, which borders Avenue Costa e Silva and has access to the city of Humaita, in Amazonas, via BR-319; the East Zone, which is limited to the Central Zone and the south with the BR-364; and, finally, the South Zone, bordering on the Central Zone (Figure 1).

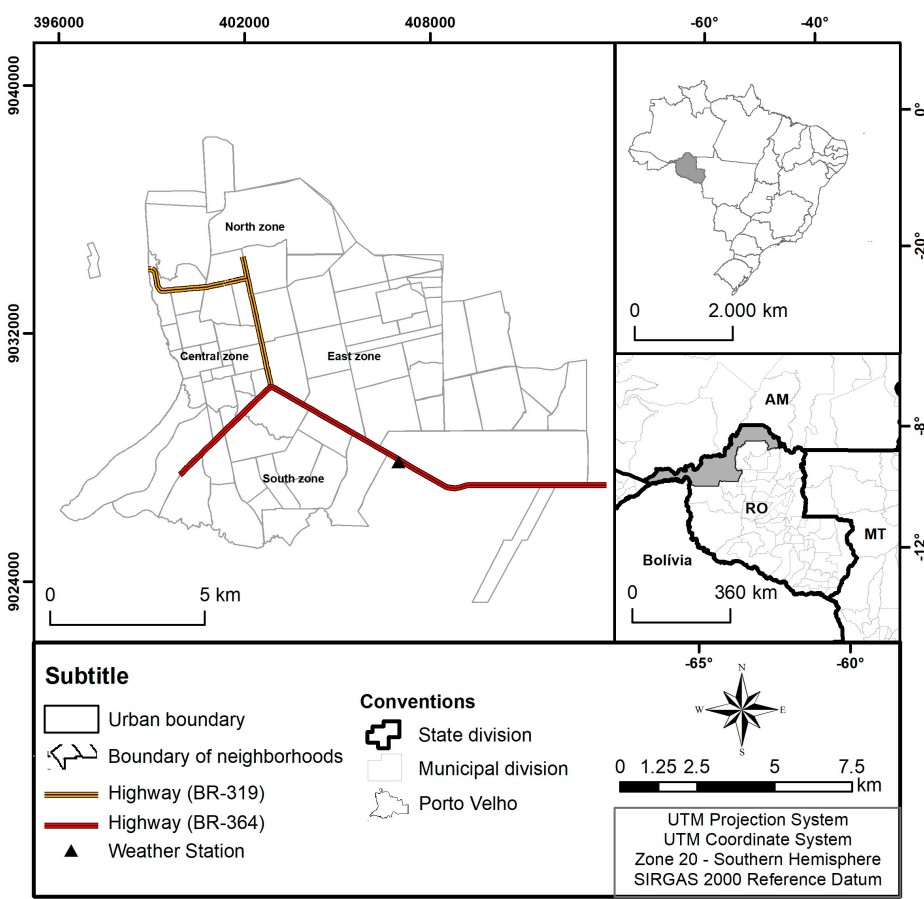

**Figure 1.** Map of the location of the urban area in Porto Velho, Rondonia. Source: organized by the authors, based on Rondonia [22,23] and Tejas [24].

Porto Velho is in a Tropical Rainy (Aw) region, according to the classification of Köppen [25–27]. It has two well-defined seasons with a short transition period between them. The rainy season occurs between November and March (southern summer) and records the highest average rainfall, 320.9 mm, in January, of the 2255.5 mm accumulated over the year [28]. The dry season (southern winter) occurs between May and September and registers the lowest monthly average of precipitation, 24.2 mm, in July, and the maximum average temperature, 34.3 °C, in August, while the minimum thermal was 18.3 °C in July [28]. Alvares et al. [26] proposed a more in-depth analysis of the Koppen climate classification, considering the regional and local levels based on temperature and rainfall data applied in the geoprocessing technique. Consequently, climate type A (tropical zone) was found for the study area, with the sub-type being AM (monsoon). The authors clarify that with the increasing fragmentation of natural landscapes and agricultural and urban uses, the hectare scale for planning seems to be the ideal approach for the future development of many applications. It is worth mentioning that since 1970, deforestation in the region has modified the space and consequently contributed to the climatic rhythm. This was demonstrated in the study by Butt, Oliveira, and Costa [29], which clarified that once this tipping point is crossed, any dieback of rain forest would result in the process of positive feedback whereby the shifting land cover causes a further reduction of the rainfall and extension of the dry season. According to the search by Coe et al. [30], deforestation, fragmentation, and conversion to agriculture create novel climates at local and continental scales because the new vegetation generally has shallower roots, a shorter growing season, lower water demand, and higher albedo than the natural vegetation it replaced.

The most recent study on climate classification by Novais and Machado [31] stands out, considered a methodological system that uses data obtained through climate reanalysis and data modeling to adjust the limits of climate units according to the adopted scale. In this study,

Porto Velho (Rondonia) is located in the Torrid (or very hot) Zone and is between 22.5 °C and 28.2 °C, with a humid equatorial climate unit with three dry months. The cartographic modeling presented by Novais and Machado [31] represented an important advance for this research, as it allowed the processing of a large volume of data in a reduced time.

### 2.2. Rhythmic Analysis and the Organization of the Climate Database

Climatic elements, such as pressure, temperature, humidity, wind, rain, and air masses, were monitored during October 2017, January, April, and August 2018 and are, thus, tabulated and organized in MS-Excel® spreadsheets, version 2013. Subsequently, these data were analyzed through the climatic rhythm to characterize the types of weather operating in the urban area, following the dynamics of regional atmospheric circulation. The results were analyzed for the dry–rainy transition season, rainy season, wet–dry transition, and dry season.

The monitoring of air masses active in the study area used images from the GOES-13 and 16 satellites. Its image collection is available free of charge on the Satellite and Environmental Systems Division website, whose electronic address is http://satelite.cptec.inpe.br/acervo/goes.formulario (acessed on 20 November 2017). To better understand the active air masses, the synoptic charts were obtained from the Navy Hydrography Center at 12 h UTC (Coordinated Universal Time), corresponding to 8 h in the local time. The analysis charts of the technical bulletins from the Center for Weather Forecasting and Climate Studies (CPTEC/INPE) were also used daily at different atmospheric levels, which helped to interpret the climatic phenomenon (http://tempo.cptec.inpe.br/boletimtecnico/pt, acessed on 20 November 2017), as well as the use of the Skew-T Log-P diagram, which describes the vertical profile of the atmosphere, used for weather forecasting (http://www.master.iag.usp.br/observados/mapa/sondagem/, acessed on 20 November 2017). Figure 2 shows the materials used to classify air masses, such as 2 October 2017 (8 a.m.) with the Tropical Continental air mass.

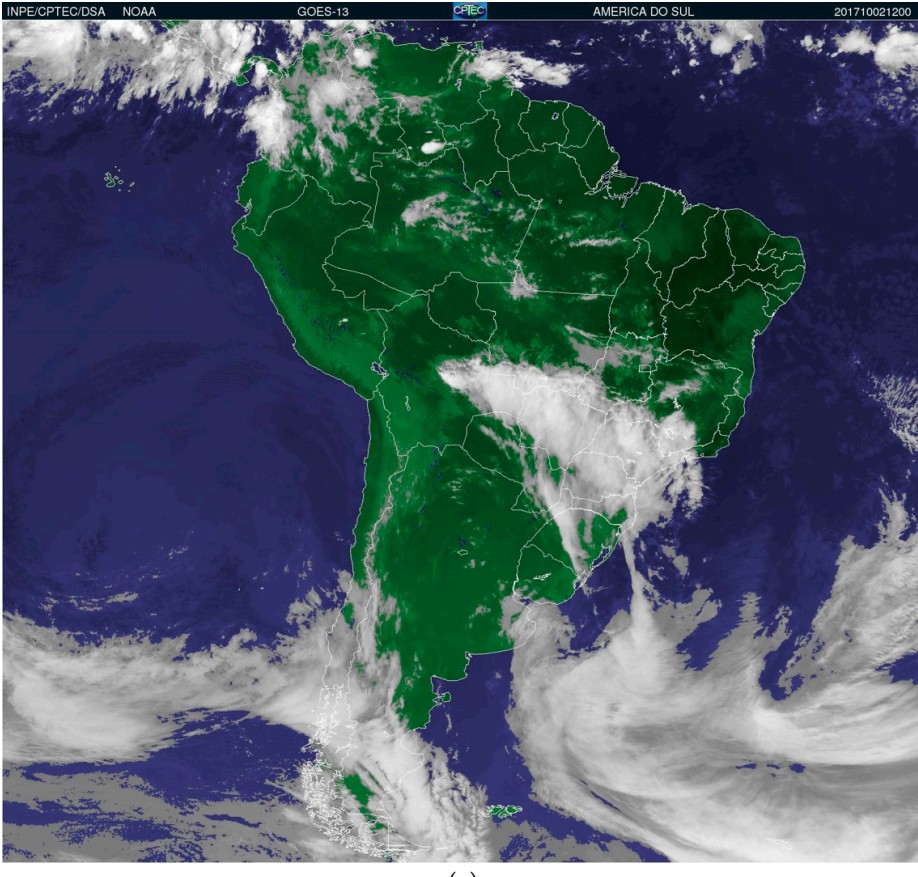

(**a**)

**Figure 2.** *Cont*.

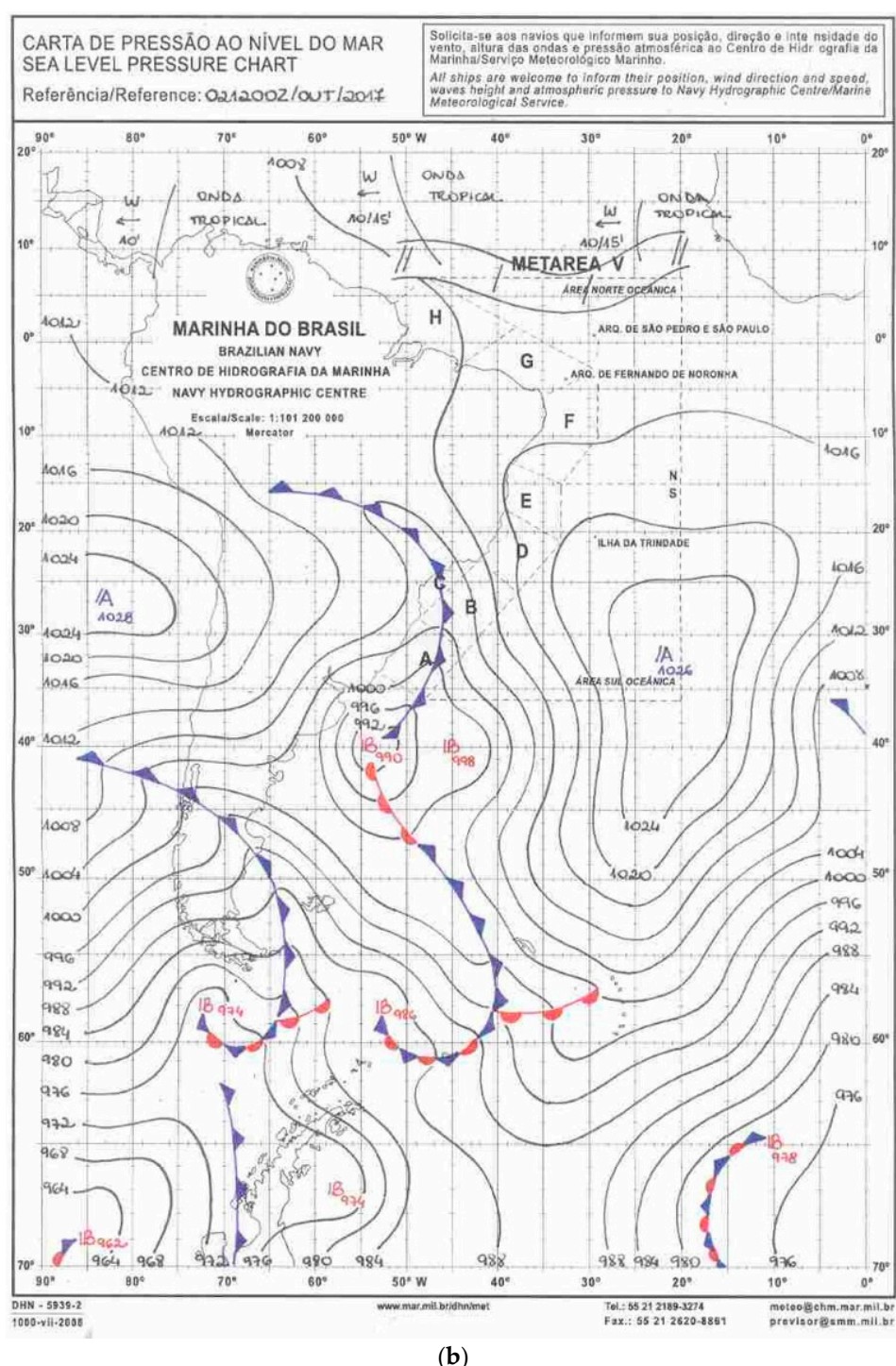

(**b**)

**Figure 2.** *Cont.*

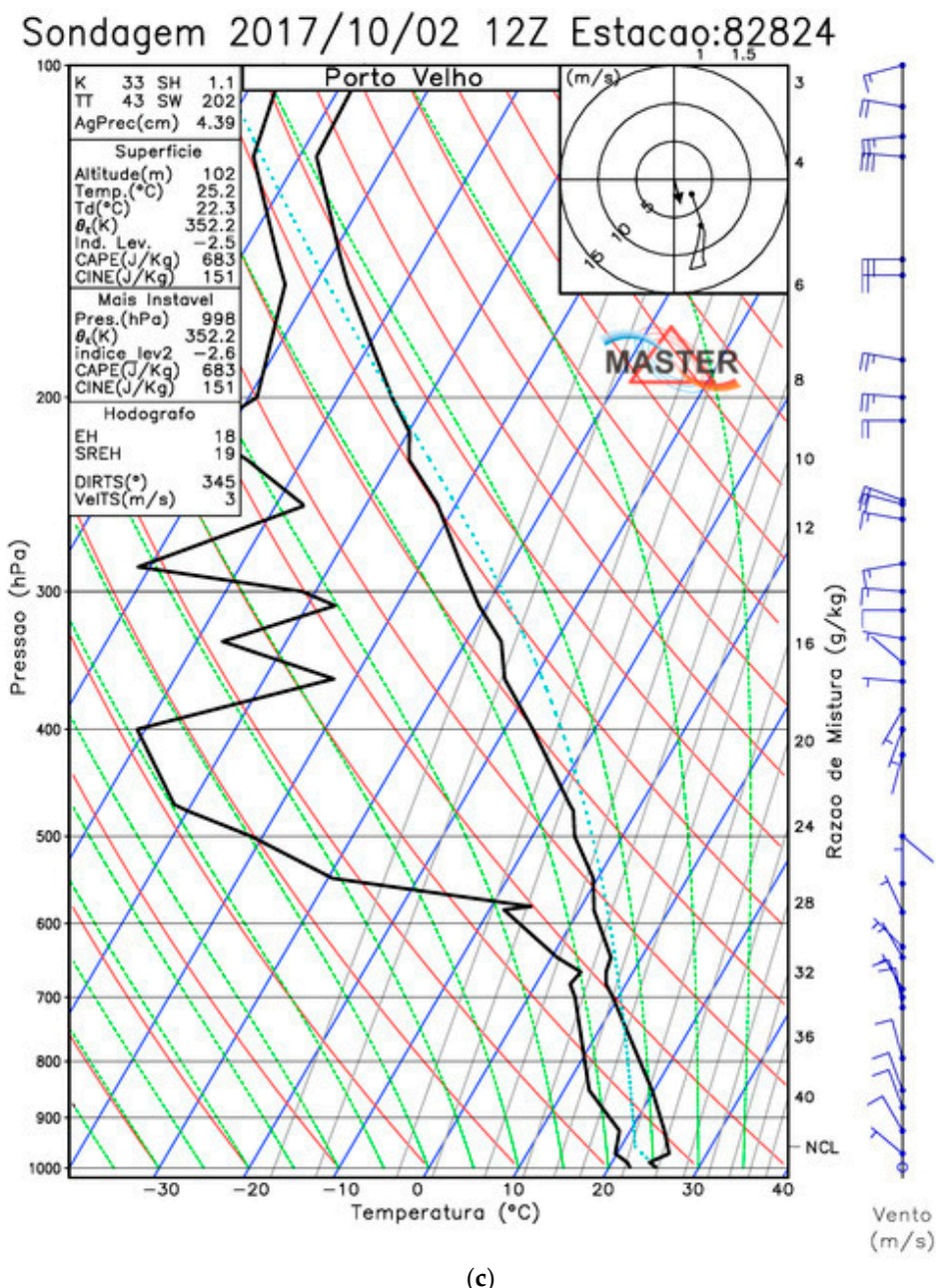

(**c**)

**Figure 2.** Satellite image (**a**), synoptic chart (**b**), and Skew-T Log-P diagram (**c**), for 2 October 2017, 8 a.m. Legend: figure (**b**) blue represents cold front, red represents warm front and when blue and red line can represent occluded or almost stationary front; figure (**c**) red represents the air temperature, blue represents the dew point temperature, green represents the plot temperature, black represents the dew point temperature line on the left and the ambient temperature line on the right. Source: organized by the authors, based on Tejas [24] and CPTEC/INPE [32].

The climate database was organized in two stages of the research. Air temperature, air humidity, wind speed and direction, atmospheric pressure, and precipitation were obtained using the INMET automatic weather station (A-925), code WMO 81932, located on the premises of the Brazilian Agricultural Research Corporation (EMBRAPA) (8°47′ S, 63°50′ W, 87 m) in Porto Velho. The data were tabulated in MS-Excel®, version 2013, followed by the graphical analysis in the free program RÍTMOANÁLISE [19] at 12 h UTC (8 h, local time) because the synoptic chart was only available at this time. The climatic data for August showed faults on some days. It was decided not to fill in the missing data, following the methodology for rhythm analysis.

In a second climate database, the wind and rain data for each day of the month were analyzed hourly and introduced in another program, the WRPLOT VIEW. The WRPLOT VIEW is a program from the company Lakes Environmental, available at www.weblakes. com/products/wrplot/index.html (acessed on 21 March 2018), to manage the direction of winds and rains through the construction of the wind and rain rose.

*2.3. Statistical Analysis*

Following this step, the Chi-Square test was performed for climatic elements (precipitation, wind speed, and direction) and atmospheric systems (air masses). This non-parametric test was chosen to statistically verify the frequency between two variables. Contingency tables were constructed for its application. First, tables were created counting the occurrences, or not, of rain in the presence of different air masses. Contingency tables for the occurrence of rain and wind direction were also created. Finally, contingency tables were created correlating wind direction and air masses. The test evaluated the degree of independence between two variables that will be analyzed, which may be dependent (alternative hypothesis—$H_1$) or independent (null hypothesis—$H_0$), with a significance level with a risk of 5% ($p$-value < 0.05), and reject to $H_0$. The test was manipulated in the programming language R Studio (R Project for Statistical Computing) through the application of libraries (dplyr; rstatix; psych; corrplot; PerformanceAnalytics; ggplot2; lubridate).

## 3. Results

*3.1. Spatial and Rhythmic Analysis of the Transition Period between "Dry and Rainy"*

Figure 3a enables the analysis of the spatial behavior of air masses in the north region in October 2017. The air masses that prevailed were Continental Tropical (mTc) air mass with 62.9% of activity, followed by Continental Equatorial (mEc) air mass with 25.8%, as well as remnants of the Maritime Tropical (mTa) air mass, 8.1%, and the Maritime Polar (mPa) air mass, 3.2% (Figure 3b).

The mTc was mainly responsible for the low precipitation rate, which did not exceed 1 mm in the analyzed period. However, the state of aridity intensifies with the high-pressure center entrance (east of the Andes and south of the tropics) associated with the formation of a depression in the Chaco region. The mPa caused the peak atmospheric pressure, 1002.2 (hPa), and a thermal minimum of 21.9 °C. Both situations occurred on 7 October 2017. The lowest atmospheric pressure values and the highest humidity and precipitation occurred due to the action of the air masses mTa and mEc (Figure 4).

The transition period between dry and rainy for Porto Velho is related to mTc (62.9%), which is fundamental in the weather and determining the climatic rhythm sustained by the low precipitation occurrence. The drop in temperature and the increase in atmospheric pressure are related to the entry of the mPa. Maximum relative humidity and rare precipitations are associated with the entrance of the mTa and the strengthening of the mEc.

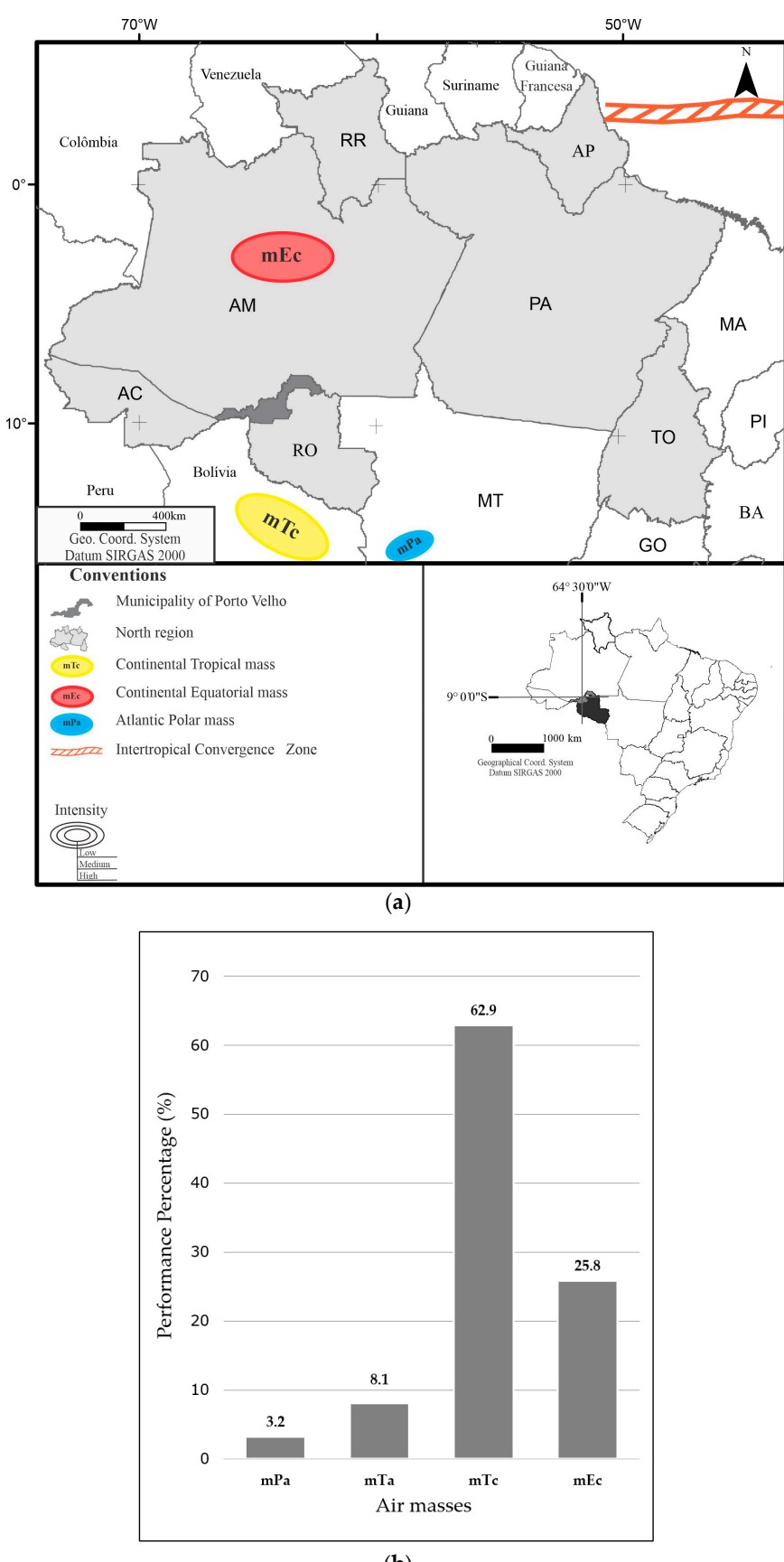

**Figure 3.** Spatialization of air masses in the north of Brazil (**a**) and percentage of activity (**b**) with direct influence on the study area during the transition between dry and rainy. Source: organized by the authors, based on Rondônia [23], IBGE [33], and CPTEC/INPE [32].

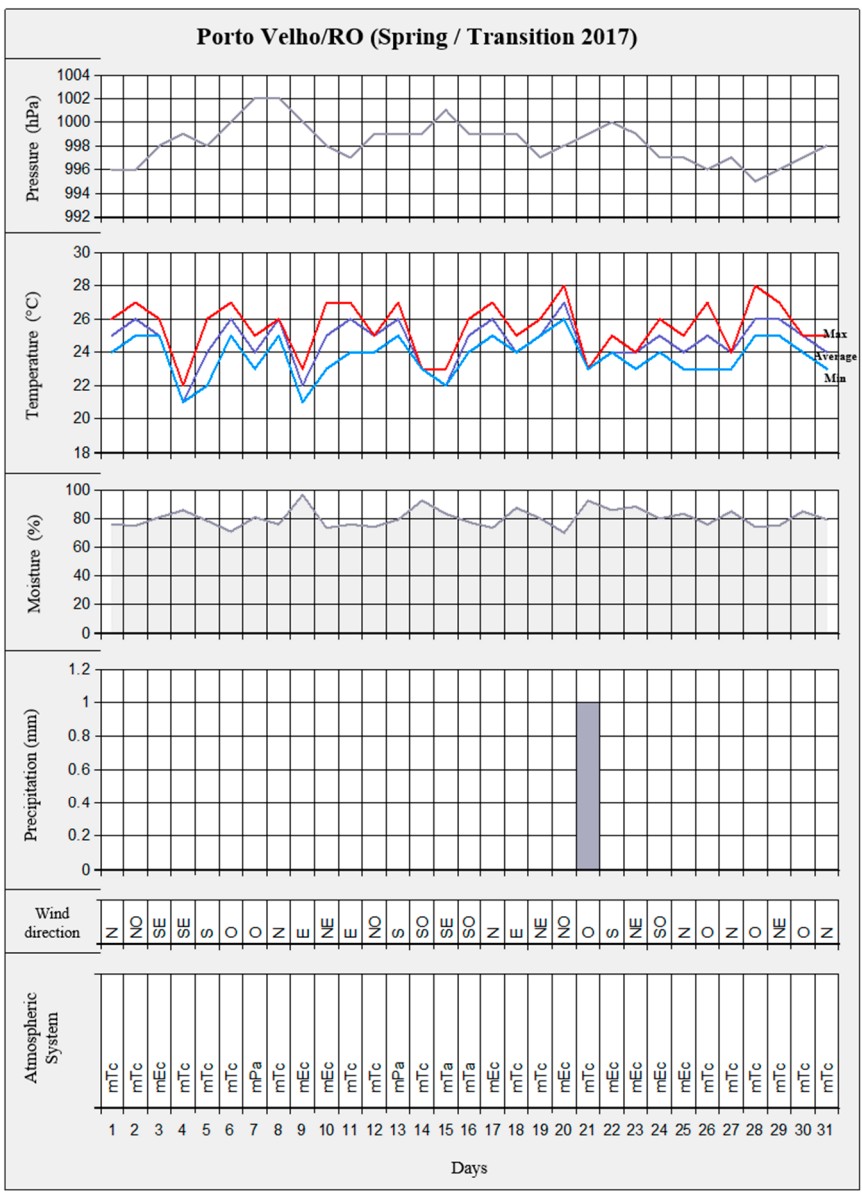

**Figure 4.** Rhythmic analysis in daily units at 8 h local time in October 2017 in Porto Velho, Rondonia. Legend: in the temperature graph, red represents maximum air temperature, dark blue represents average air temperature and light blue represents minimum air temperature. Source: organized by the authors, based on Tejas [24], CPTEC/INPE [32], and INMET [34].

For winds, 38.6% of the reports did not present data, and for rainfall, 38.9% were also without records. The predominant wind direction was south and southwest (Figure 5a), reaching a speed between 0.50 and 2.10 m/s, reaching 41.7% of light breezes (Figure 5b). The winds' low intensity is related to the formation of frontal systems in the Brazilian central-southern region and the influence of mTc on the study area. Winds above 3.60 m/s are related to the arrival of mPa. The rainfall index, in turn, presented an accumulated total of 78.8 mm. It was 92 mm less than those recorded by Tejas et al. [35] and 113.8 mm less than that measured by Brazil [28] for October. It can be explained by 67.1% of dry days due to the action of mTc, the low intensity of rainy days (0.50 to 2.10 mm/h), 1.7%, related to the strengthening of mEc, and it often being oriented to the north (Figure 5c,d).

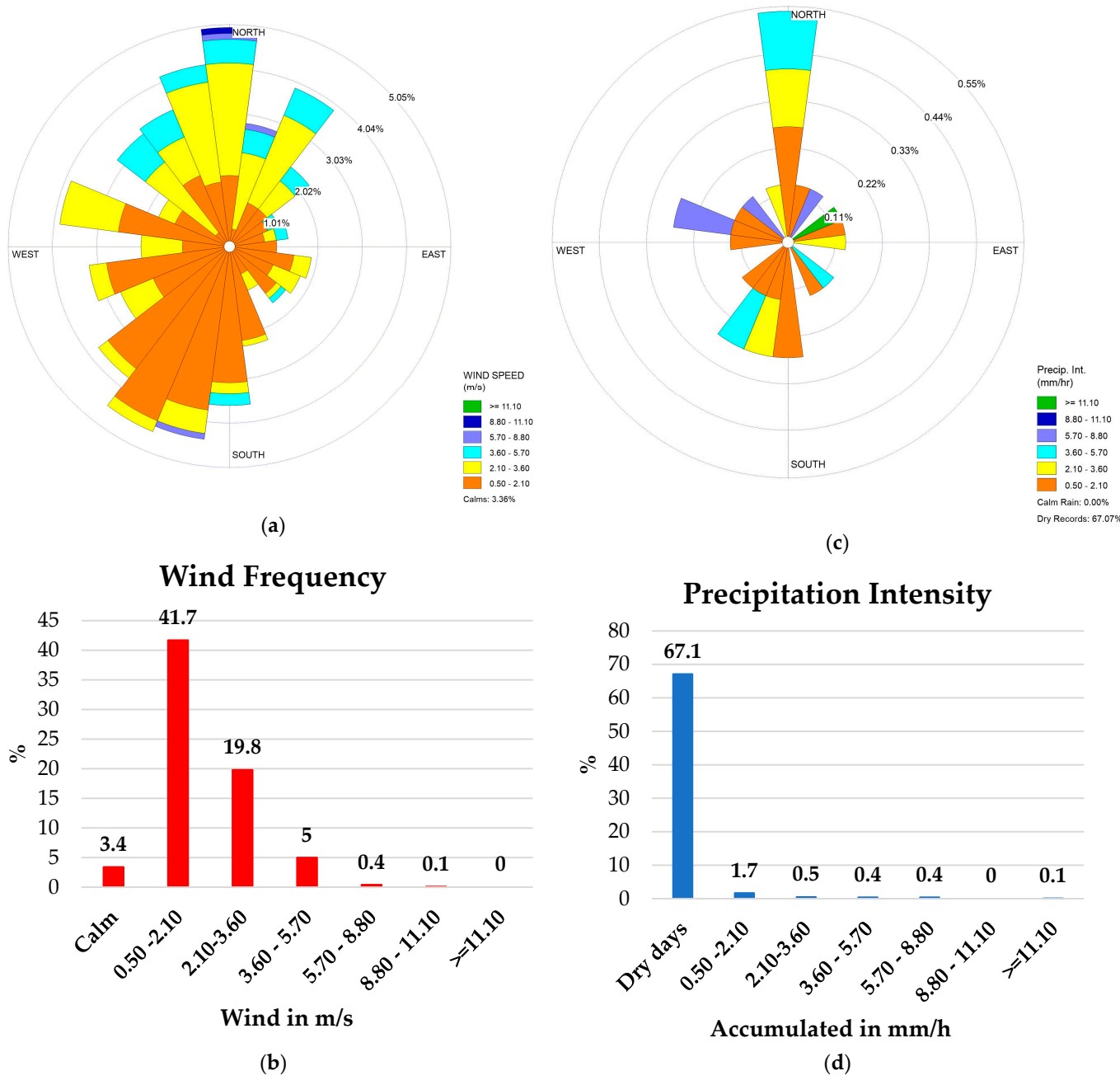

**Figure 5.** Wind rise (**a**) and frequency and (**b**) rain rise (**c**) and intensity (**d**) for October 2017 in Porto Velho, Rondonia. Source: organized by the authors, based on Tejas [24], INMET [34], and Lakes Environment [36].

Figure 6a presents the correlogram of the time series studied, in which there was a weak and positive correlation between precipitation and relative air humidity (r = 0.33). Precipitation and average temperature showed a moderate and negative correlation (r = −0.36). Average temperature and humidity showed a strong negative correlation at the 0.001 significance level (r = −0.81). Also, Figure 6b,c presents the behavior and frequency in (d,e) of the rain and wind variables for the transition period (October 2017). When the non-parametric Chi-Square test was applied to these variables in the association between the air masses' action and the rain occurrence, the results presented a *p*-value of 0.86, with three degrees of freedom and an $X^2$ of 0.75. The significance level adopted for these results was 5%. Thus, the null hypothesis ($H_0$) is not rejected because the variables are not independent for this analyzed period.

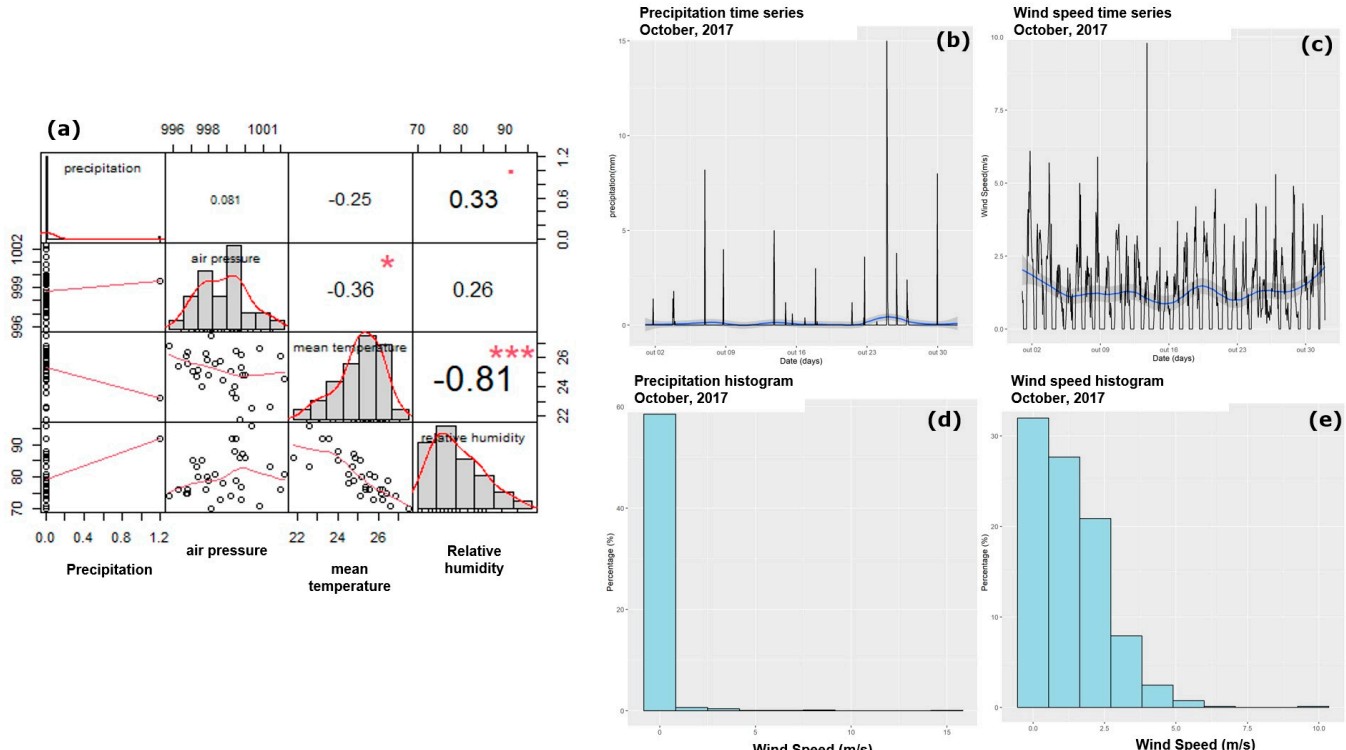

**Figure 6.** Brief exploratory analysis using the graphs (**a**) correlogram, (**b**) precipitation/day/month, (**c**) precipitation and wind speed/day/month, (**d**) precipitation/wind speed histogram, and (**e**) wind speed histogram for October 2017. Legend: *** Significant correlation at the 0.001 level; * Significant correlation at the 0.05 level. Source: organized by the authors.

When analyzing the relationship between wind direction and precipitation data, it was concluded that the null hypothesis ($H_0$) should not be rejected since the data obtained by the test revealed a *p*-value (0.744), with seven degrees of freedom and $X^2$ (4.3056). Also, Figure 6 shows the frequency of precipitation (>50%) with low wind speed in (c), as well as the distribution of wind speed for October, with (>20%) between 0 and 2.5 m/s (d).

### 3.2. Rhythm Spatial Analysis of the Rainy Period

The rainy season (southern summer) occurs due to the strengthening of the mEc (85.5%) and, at times, due to the influence of the South Atlantic Convergence Zone (SACZ), as can be seen in Figure 7a,b.

The incidence of mEc and the formation of SACZ and Bolivian High (AB) influenced the climatic elements during January 2018, which can be seen in the graph of rhythmic analysis (Figure 8). In January 2018, the atmospheric pressure fluctuated between 989.3 hPa and 996.9 hPa. The lowest values reflect the mEc performance, and the highest values are related to mTc in conjunction with Bolivian High (BH) and the South Atlantic Convergence Zone (SACZ).

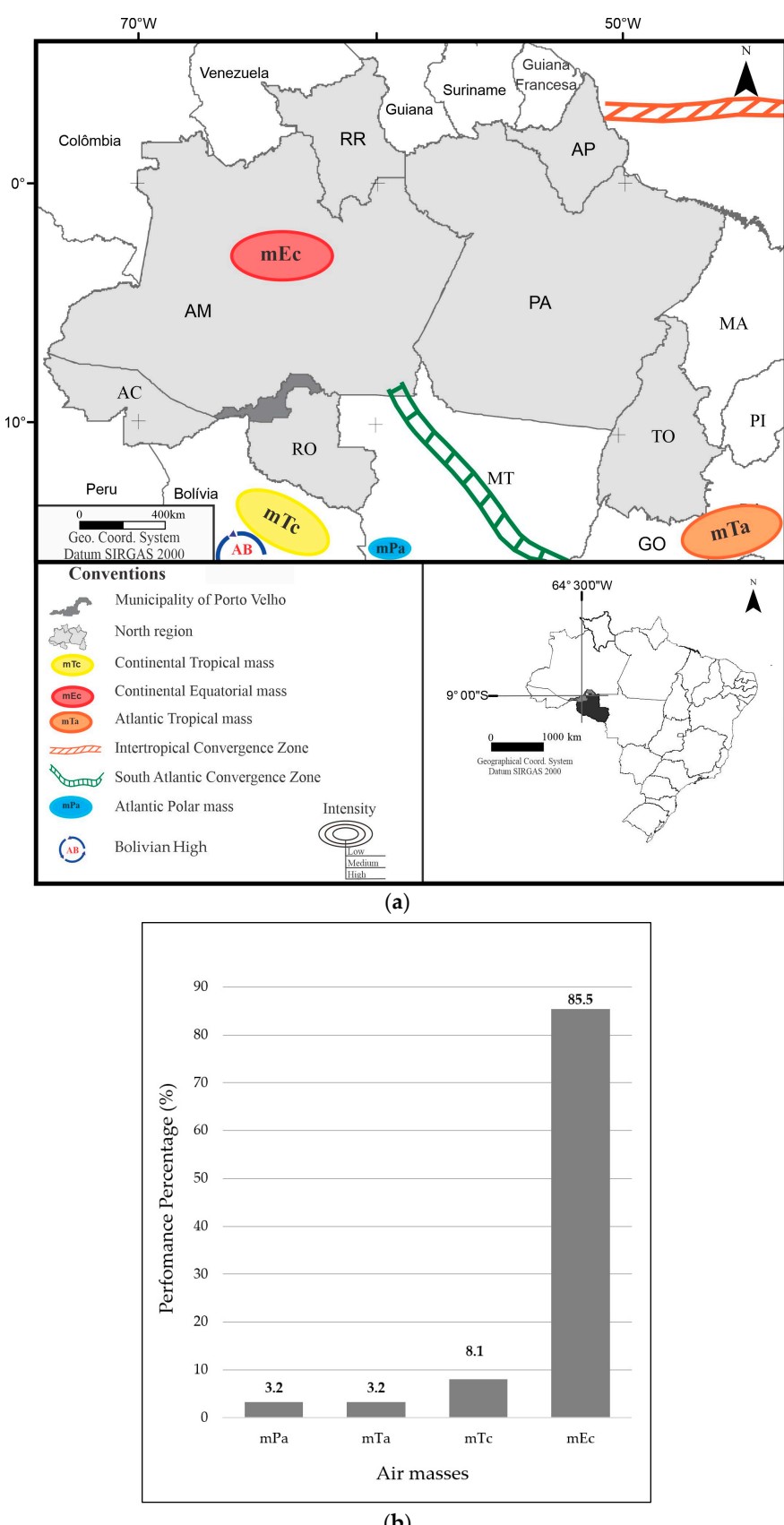

**Figure 7.** Spatialization of air masses in the north of Brazil (**a**) and percentage of activity (**b**) with direct influence on the study area in January 2018. Source: organized by the authors, based on Tejas [24], Rondonia [23], IBGE [32], and CPTEC/INPE [37].

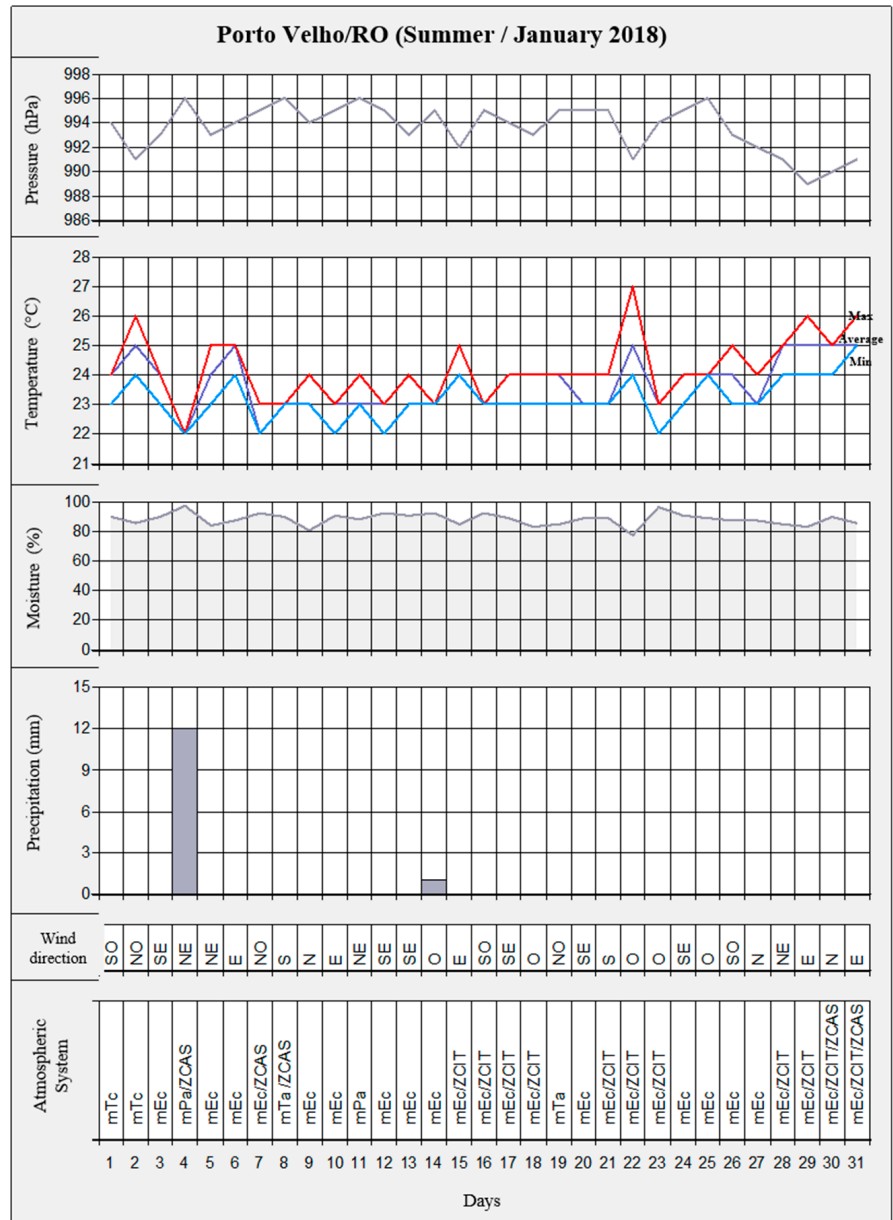

**Figure 8.** Rhythmic analysis in daily units at 8 h local time in January 2018 in Porto Velho, Rondonia (rainy season). Legend: in the temperature graph, red represents maximum air temperature, dark blue represents average air temperature and light blue represents minimum air temperature. Source: organized based on Tejas [24], CPTEC/INPE [37], and INMET [34].

Three atmospheric circulation situations must occur for the formation of a SACZ, the first related to cyclogenetic events in the South Atlantic Ocean; the second related to the advance of a classic cold front over Southeast Brazil; and the third, an anticyclone at high levels (AB). When these systems are organized, a band of cloudiness forms in the south of the Amazon region up to the South-Central Atlantic for at least four days (it can persist for up to 10 days) [6,38,39].

The atmospheric circulation caused rains of 12 mm at 8 h on the 4th and a drop in the air temperature, reaching the value of 22.3 °C (Figure 8). Santos Neto et al. [40] highlight that the behavior of precipitation is due to the reflection of the west and east wind regimes in low levels of the atmosphere at this moment of the year. The western regimes are associated with SACZ, as are the Low-Level Jets, while the eastern regimes are associated with local convection and instability lines.

Another highlight of the climatic behavior in the southern summer is the performance of mEc (85.5%), responsible for the thermal maximums and the peaks of air humidity in January 2018. Such as on the 22nd, when the thermometers recorded a maximum of 27.3 °C, and the humidity reached 97%. On that day, there was an action by SACZ.

Hourly data from the INMET wind weather station (Figure 9a,b) revealed a light breeze in January: 70.2% did not exceed 2.10 m/s, with a predominant orientation from north to south. From the middle to the end of the month, winds reached speeds between 3.6 and 5.7 m/s (weak to moderate), and the predominance of the direction changed from northeast to northwest. This demonstrates the influence of SACZ on the orientation of local winds.

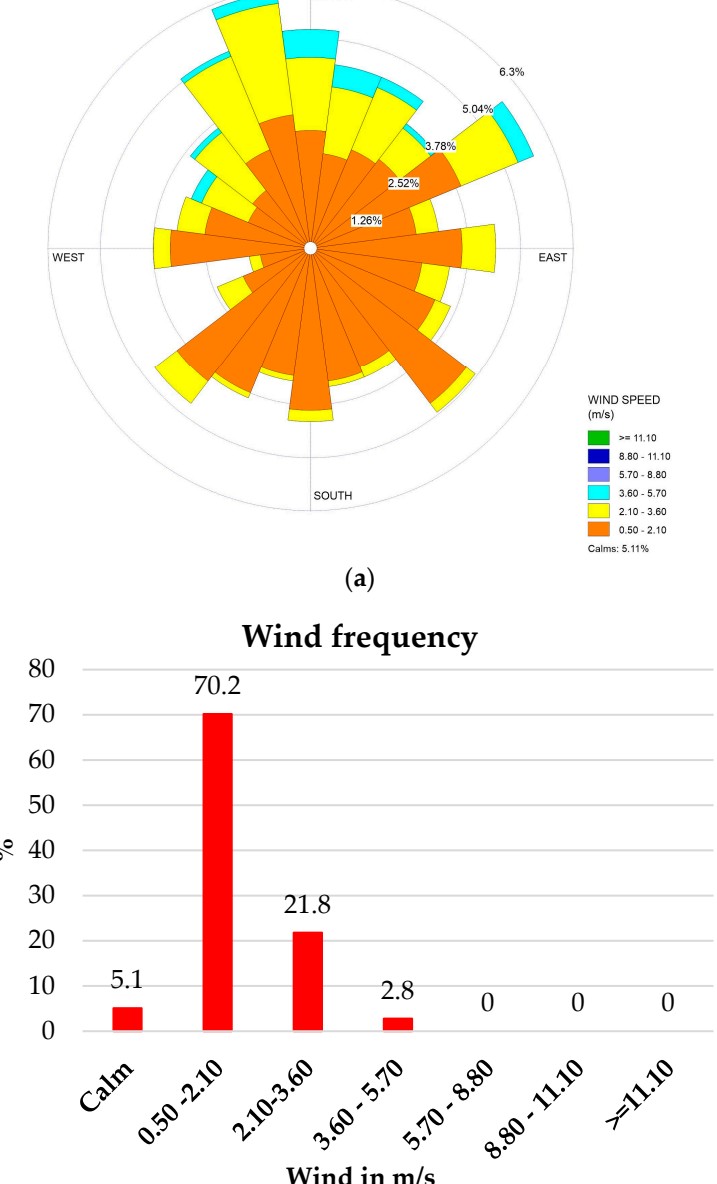

Figure 9. *Cont.*

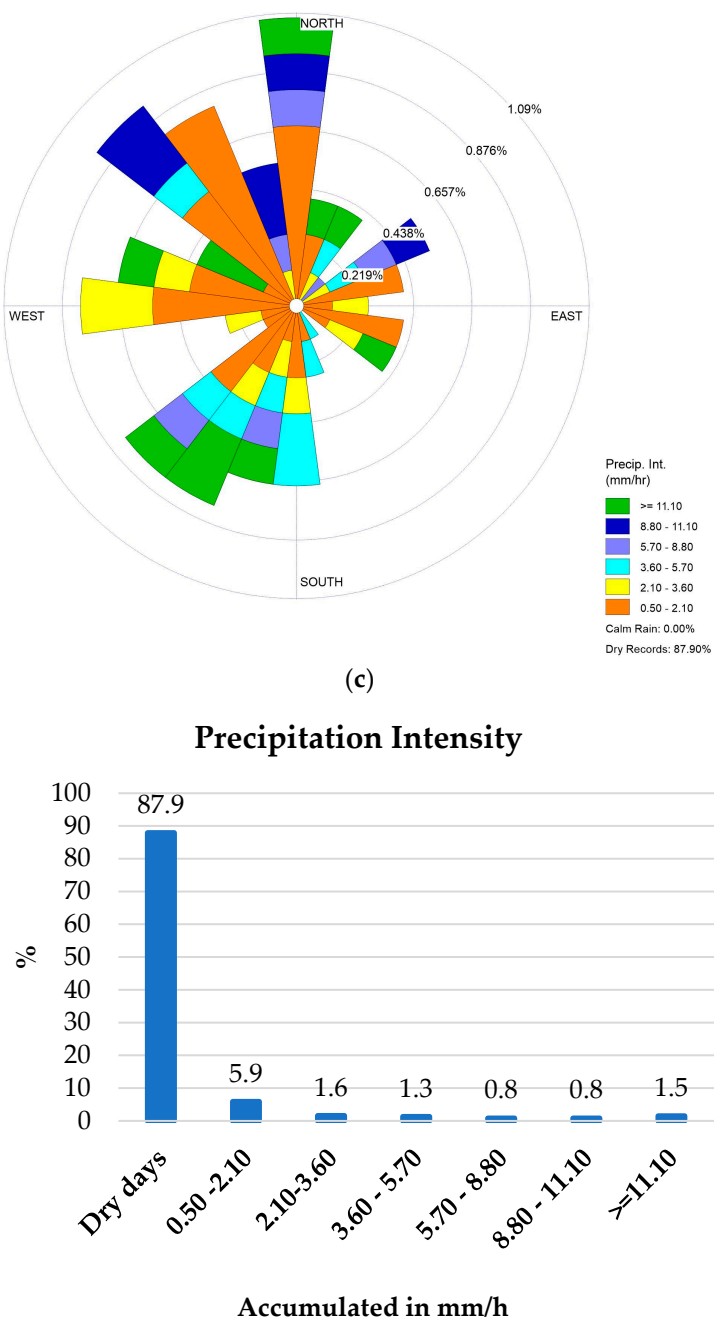

(**c**)

## Precipitation Intensity

(**d**)

**Figure 9.** Wind rise (**a**) and frequency (**b**) and rain rise (**c**) and intensity (**d**) for January 2018 in Porto Velho, Rondonia. Source: elaborated by Tejas [24], INMET [34], and Lakes Environment [36].

Precipitation in January 2018 amounted to an accumulated total of 443 mm. It was 122 mm higher than that monitored by INMET in the 1961–1990 Normal Climatological (Brazil 1992) and 100 mm in the accumulated recorded for the month in the study of climatic variability between 1999 and 2011 by Tejas et al. [35]. However, 88% of the days were dry, and only 1.6 had volumes higher than 2.1 mm/h (Figure 9c,d). Therefore, it indicates that the rains are voluminous and increasingly concentrated in some hours of the day. In the case of the rain on the 4th, there is a direct influence of the formation of a cloud band caused by the SACZ. The record of rain in a single day had consequences in the urban environment of Porto Velho for the formation of overflow and floods of the streams in the city.

In the statistical analysis of January 2018, considered the rainy season of the study area, represented by Figure 10a, a moderate and positive relationship (r = 0.42) is observed in

the correlogram of precipitation and relative air humidity, and humidity was also active in the study area when rain occurred. For the parameter's atmospheric pressure and average temperature, there was a strong negative correlation (r = −0.79) with a significance level of 0.001. In the mean air temperature and humidity, the result was r = −0.75, a strong and negative relationship with a significance level of 0.001.

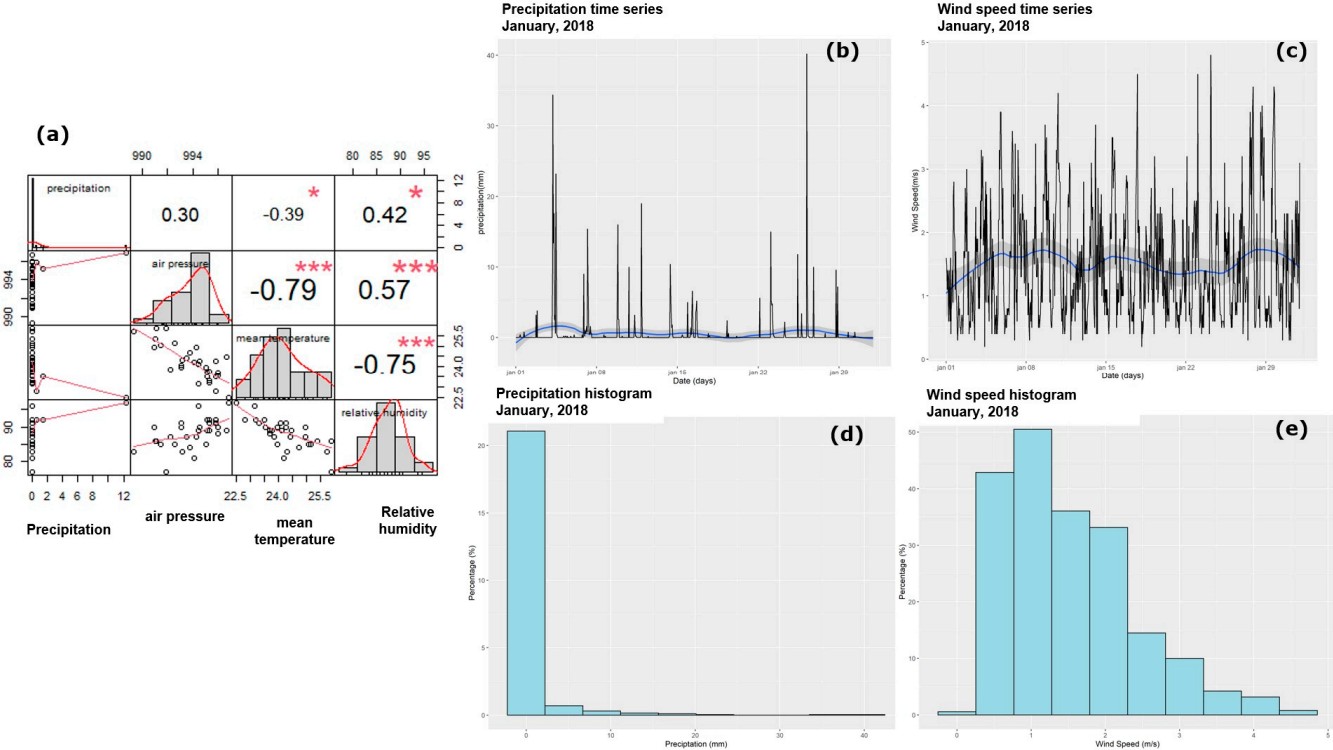

**Figure 10.** Brief exploratory analysis using the graphs (**a**) correlogram, (**b**) precipitation/day/month, (**c**) precipitation and wind speed/day/month, (**d**) precipitation/wind speed histogram, and (**e**) wind speed histogram for January 2018. Legend: *** Significant correlation at the 0.001 level; * Significant correlation at the 0.05 level. Source: organized by the authors.

In Figure 10b, the results of air masses and precipitation parameters pointed to a *p*-value of 0.04, with seven degrees of freedom and an $X^2$ of 14.72. Thus, a significant association was observed between the air mass variables (mEc, mTa) and rainfall occurrences.

Figure 10c shows that the variables wind direction and rainfall were not associated. The result was a *p*-value (0.7868), with six degrees of freedom and an $X^2$ of 3.1731. Therefore, the null hypothesis ($H_0$) is not rejected. In Figure 10d,e, the frequency of precipitation was (>20%), and the wind speed distribution for January 2018 was 50% for 1 m/s. The values found revealed a statistically significant association from the *p*-value (0.02223), with 49 degrees of freedom and an $X^2$ of 70.85, to determine an association between wind direction and air masses.

### 3.3. Spatial and Rhythmic Analysis of the Transition Period between "Rainy and Dry"

The second transition period (between rainy and dry) was carried out in April 2018, with the specialized atmospheric weather, as shown in Figure 11a. The Southern hemisphere was under the southern autumn season. During this month, it was observed that mEc had its dominance reduced compared to January, when it prevailed with 86%, but still presents an important performance with 56.7% in the participation of days (Figure 11b). This low-pressure center, which is commonly formed at this time of year over the Western Amazon, promotes atmospheric instabilities, mainly due to the encounter with mTc, mTa, and mPa, which, respectively, were present in 18.3%, 18.3%, and 6.7% in April 2018.

Atmospheric pressure during April 2018 fluctuated between 991.5 hPa and 997.6 hPa, as shown by the rhythmic analysis graph (Figure 12). The lowest values occurred when there were no high-pressure systems, and mEc was strengthened. The highest values, on the other hand, occurred when mTc, mPa, and mTa operated. The minimum temperatures were reached when the mEc, with the action of the Intertropical Convergence Zone (ITCZ), caused the formation of cloudiness, bringing the air humidity close to condensation. Also, this occurs when the third branch of the mPa advances through the Amazon region, which happened on the 9th day when the thermometers registered 22.8 °C. High air temperatures were recorded under the effect of mTc. Finally, the humidity varied from 82% to 97%. The lowest indexes were related to the entrance of mTc and mPa, and the highest indexes occurred when the mEc was strengthened.

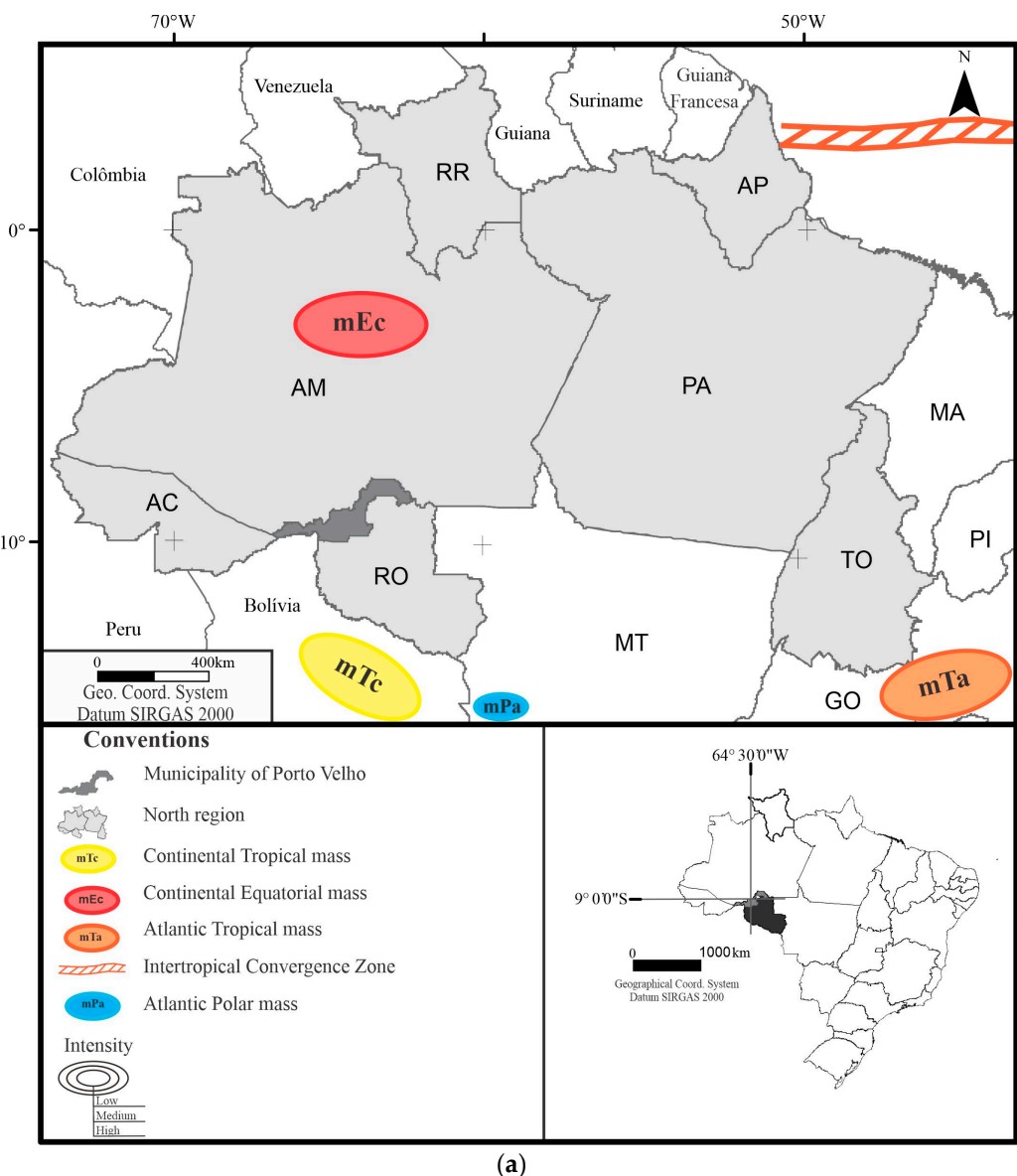

(**a**)

**Figure 11.** *Cont.*

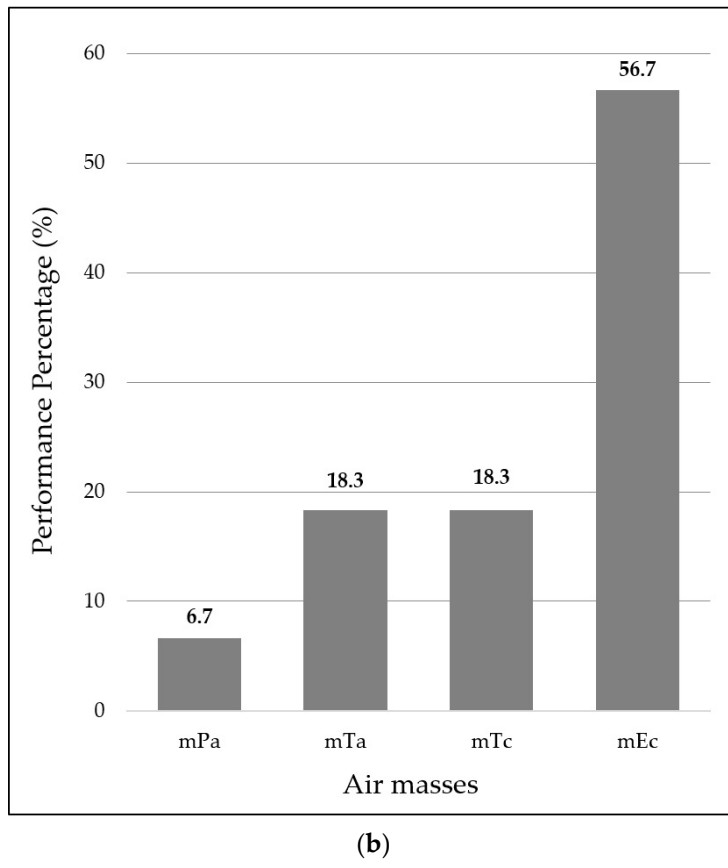

(**b**)

**Figure 11.** Spatialization of air masses in the north of Brazil (**a**) and percentage of activity (**b**) with direct influence on the study area in April 2018 in Porto Velho, Rondonia. Source: organized by the authors, based on Tejas [24], CPTEC/INPE [37], and CHM [41].

The winds (with 72.4%) had a predominant speed of up to 2 m/s, often oriented from south to southeast (Figure 13a,b). Regarding precipitation, 91.4% of the days were arid and in a diffuse direction (Figure 13c,d). On 16 April, there was precipitation of 30.8 mm due to the meeting of the mPa with the mEc. The accumulated total for this month was 238 mm. A superior index at 159.8 mm, when compared to the transition period between dry and rainy, is lower, respectively, by 13 mm and 8 mm of the rainfall values for April, monitored by INMET [34] and Tejas et al. [35].

Figure 14a shows the level of correlation between moderate and negative climatological variables between temperature and air humidity (r = −0.57), with a significance level of 0.01. As it was a transition season, the temperature was high, and the air humidity value was low.

The proposed application of the Chi-Square test regarding the independence of air masses and precipitation for April 2018 resulted in a *p*-value of 0.51, with 4 degrees of freedom and $X^2$ (3.31). Therefore, the null hypothesis ($H_0$) is not rejected for this period of air masses action, specifically the mEc, with the occurrence of precipitation. Figure 14b enables the observation of this parameter.

When analyzing wind direction and speed with rainfall data from the Chi-Square test, the results are *p*-value (0.79), with 6 degrees of freedom and $X^2$ (3.17). Therefore, the null hypothesis ($H_0$) cannot be rejected. Figure 14d,e shows the frequency of precipitation (45%) with low wind speed and the distribution of wind speed for April, with 60% between 0 and 2 m/s.

However, it was decided to analyze the degree of independence between the air masses and wind direction variables. The values obtained through the Chi-Square test resulted in a *p*-value of (0.68), with 24 degrees of freedom and $X^2$ of 20.18. Therefore, this result

expresses the rejection for $H_0$, although there is an association between air masses/wind direction for mEc, mPa, mTa, and mTc in the study area.

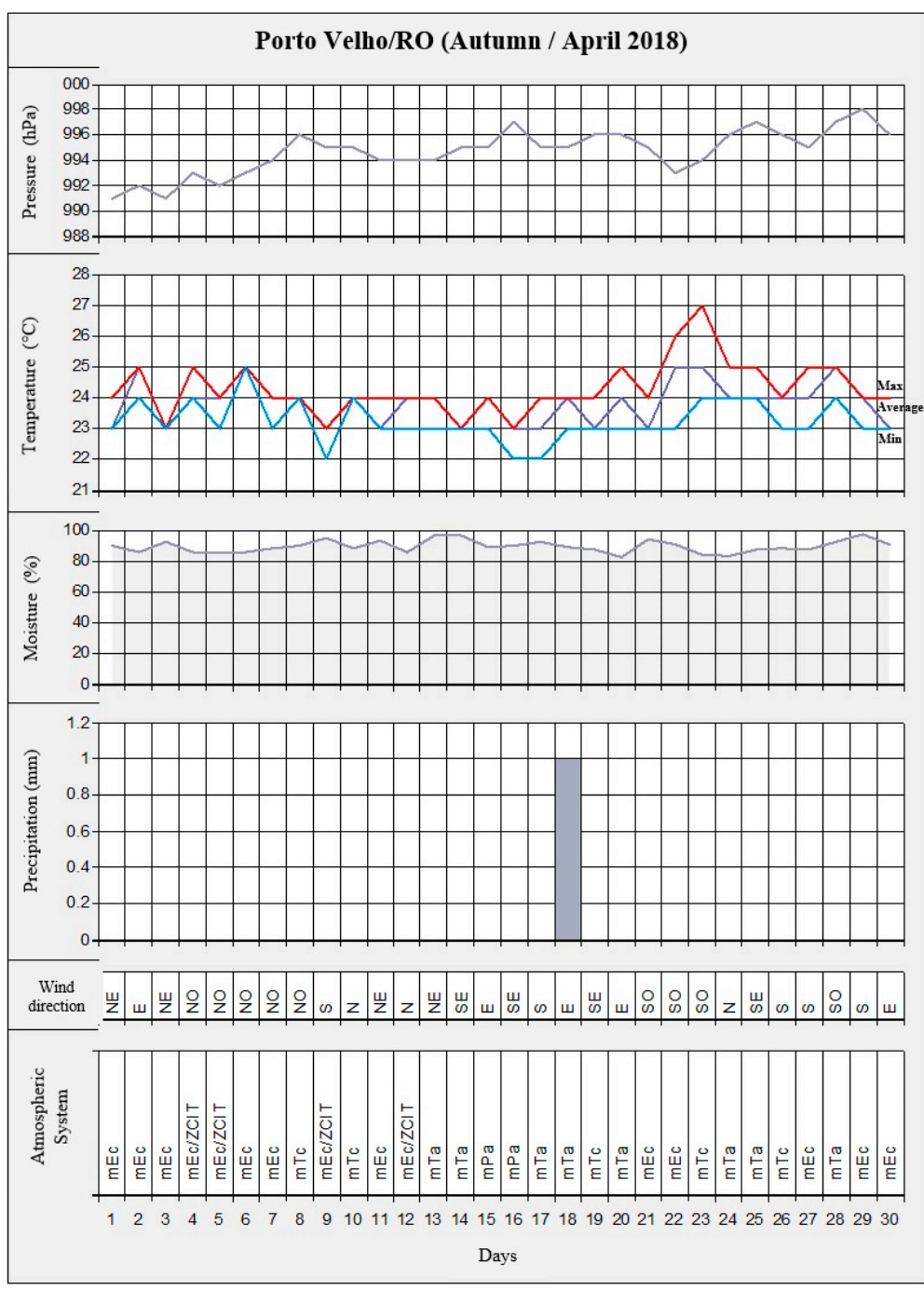

**Figure 12.** Rhythmic analysis in daily units at 8 h local time to April 2018 in Porto Velho, Rondonia. Legend: in the temperature graph, red represents maximum air temperature, dark blue represents average air temperature and light blue represents minimum air temperature. Source: organized by the authors, based on Tejas [24], CPTEC/INPE [37], and INMET [34].

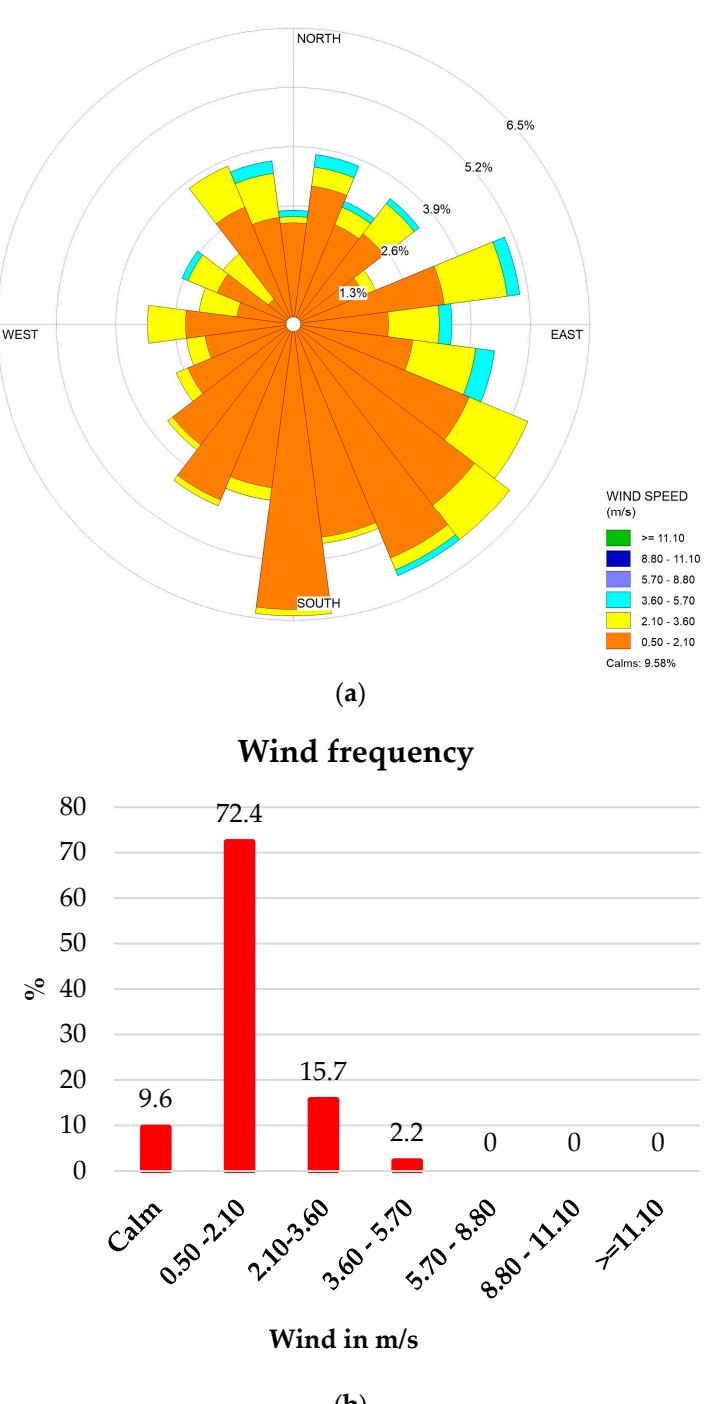

(**a**)

(**b**)

**Figure 13.** *Cont.*

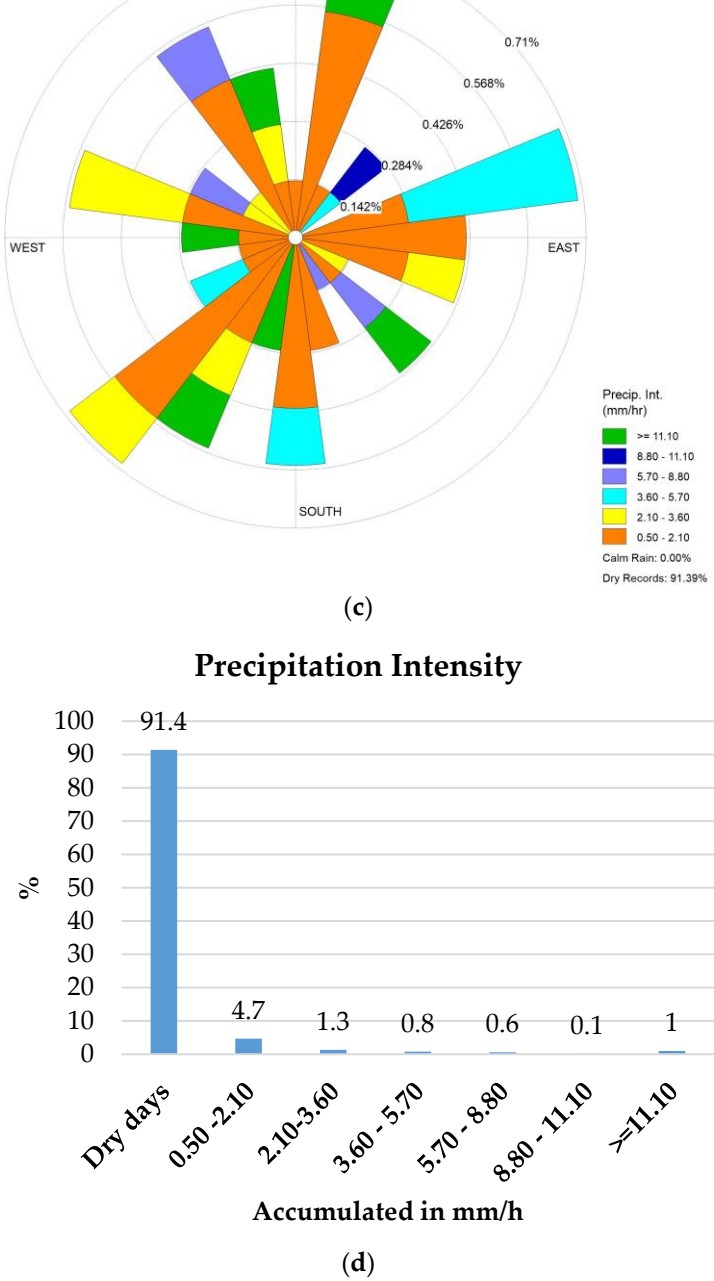

(**c**)

## Precipitation Intensity

(**d**)

**Figure 13.** Wind rise (**a**) and frequency (**b**) and rain rise (**c**) and intensity (**d**) for April 2018 in Porto Velho, Rondonia. Source: Organized by the authors, based on Tejas [24], INMET [34], and Lakes Environment [36].

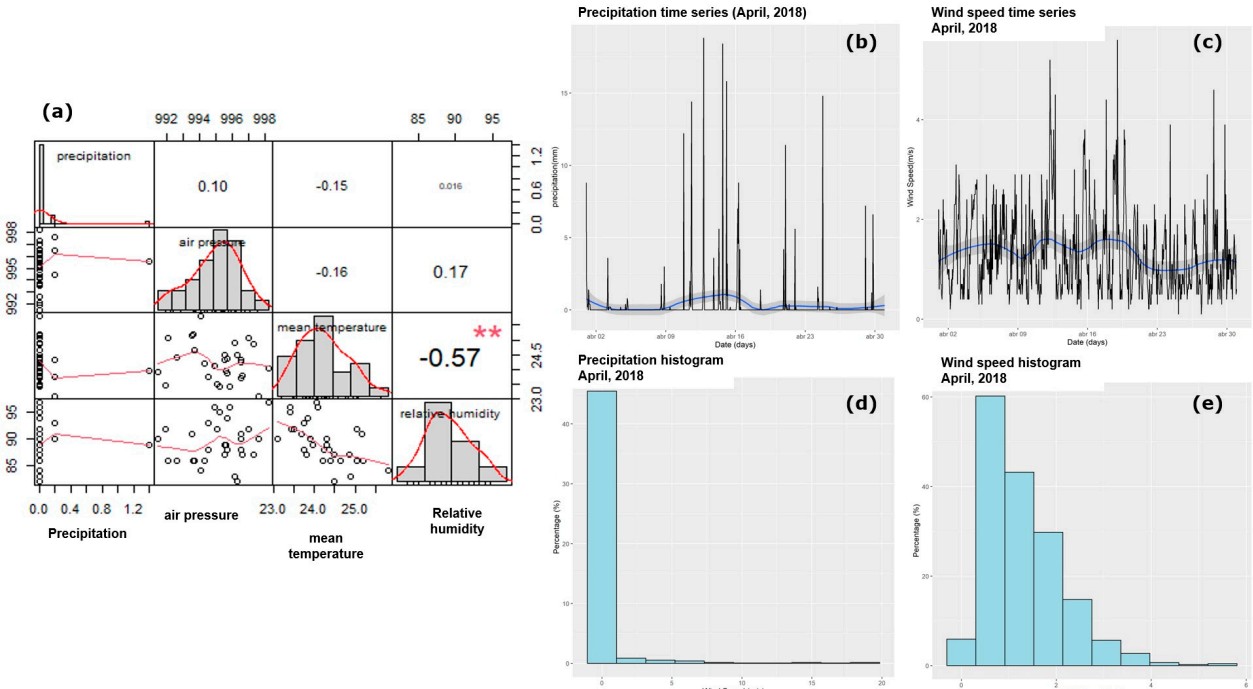

**Figure 14.** Brief exploratory analysis using the graphs (**a**) correlogram (**b**) precipitation/day/month (**c**) precipitation and wind speed/day/month (**d**) precipitation/wind speed histogram, and (**e**) wind speed histogram wind for the month of April 2018. Legend: ** Significant correlation at the 0.01 level. Source: Organized by the authors.

### 3.4. Spatial and Rhythmic Analysis of the Dry Period

The spatial analysis of air masses for the dry period occurred in August 2018, when the southern hemisphere was under the southern winter season (Figure 15a), characterized by the absence of rainfall, low relative humidity, and high air temperatures. These conditions in August 2018 were related to the action of mTa (27.4%), mTc (19.4%), and mPa (17.7%) (Figure 15b). However, mEc, with 35.5% presence, was especially important, given its role in maintaining the relative humidity above 70%. It is also noteworthy the significant increase in the performance of mPa in the region. Because when they cross the branches of penetration and combine with a tropical air mass, cold fronts are formed in the Western Amazon.

In August 2018, the atmospheric pressure ranged from 1002.1 hPa to 1009.3 hPa (Figure 16). The minimum value occurred when the mTa reached the Amazon region and found the mEc weakened (hence, with low humidity). The maximum value occurs when the mPa reaches the region preceded by the weakened mEc. It is observed that low humidity enhances the performance of high-pressure centers. The minimum temperature was 17.4 °C on the 27th due to the advance of a frontal system over the region. The maximum thermal value was 27.4 °C on day 14, under the influence of mTc. The reduced performance of mEc provided the absence of precipitation, as well as the entry of mPa and mainly mTc, was responsible for the lowest humidity indexes recorded for a one year, reaching a minimum of 67% on day 20, as a result of cold and dry air that entered the South of the Amazon. The positive peak, 90%, only occurs when the high-pressure centers are weakened and not operating in the Amazon region. The mEc was strengthened due to the high temperatures caused by the action of mTc from the 12th to the 14th.

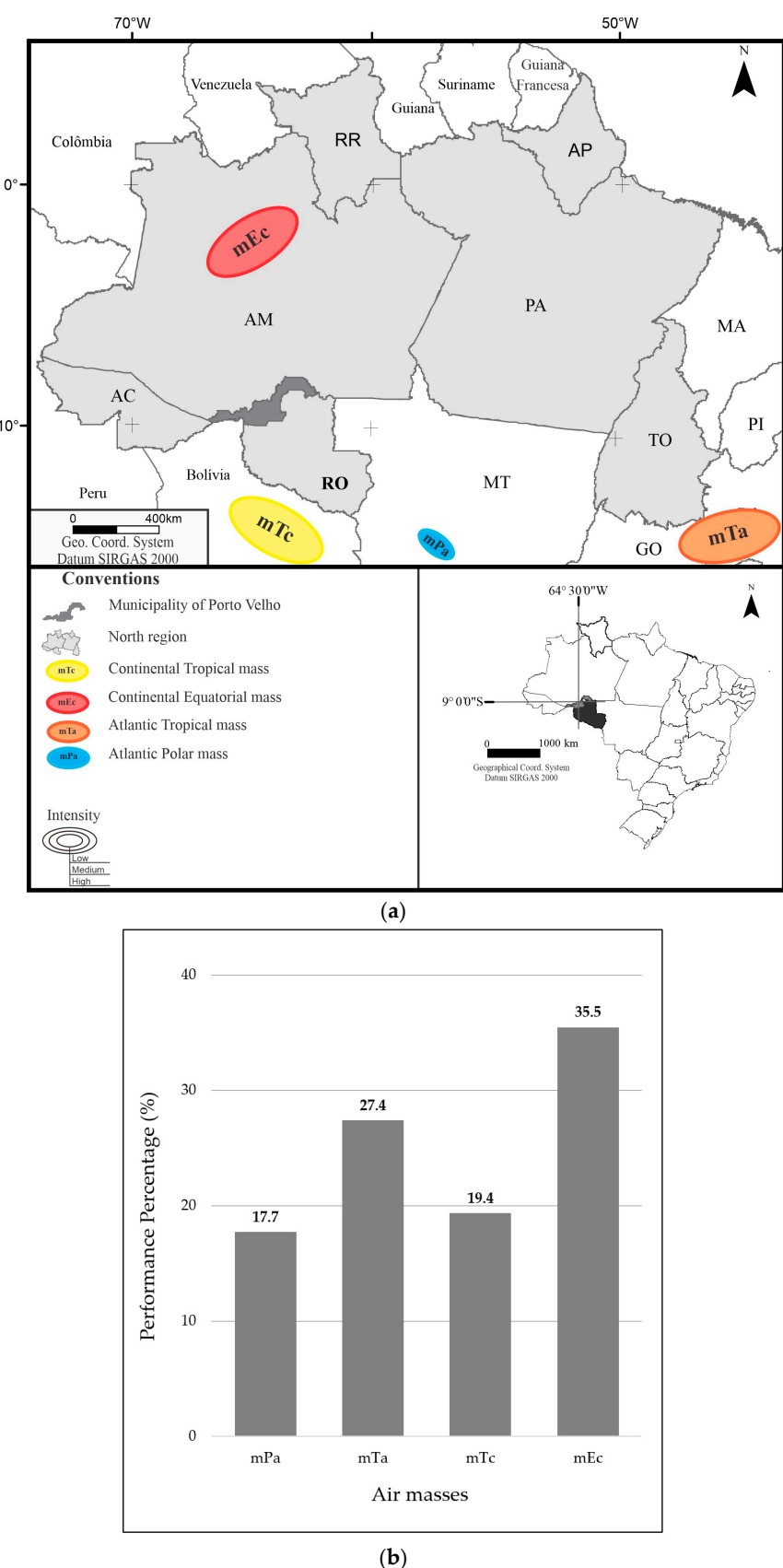

**Figure 15.** Spatialization of air masses in the North of Brazil (**a**) and percentage of activity (**b**) with direct influence on the study area in August 2018. Source: Organized by the authors, based on Tejas [24] and CPTEC/INPE [37].

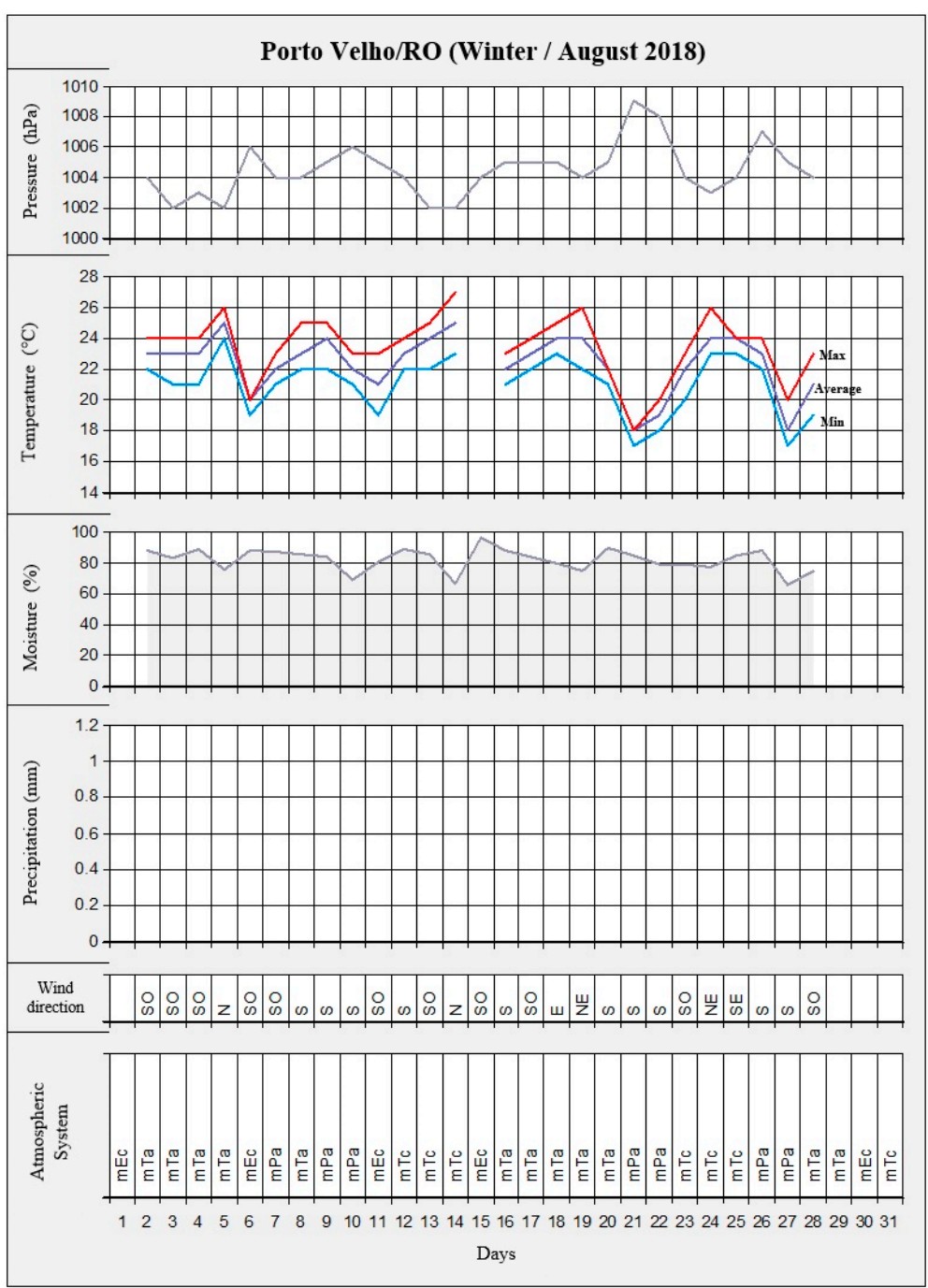

**Figure 16.** Rhythmic analysis in daily units from 8 h local time to August 2018 in Porto Velho, Rondonia. Legend: in the temperature graph, red represents maximum air temperature, dark blue represents average air temperature and light blue represents minimum air temperature. Source: Organized by the authors, based on Tejas [24], CPTEC/INPE [37], and INMET [34].

The winds had a predominance of up to 2.10 m/s, 36.2% influenced mainly by the performance of high-pressure systems. Less than 2.10 m/s, most of them were in concomitance with the mTa. Those above 2.10 m/s, most of them were in concomitance with mPa. The direction varied from southeast to southwest. The rains had an accumulated total for August 2018 of 76.8 mm, a value higher than 40.4 mm if compared to the data from "Normal Climatologica" [28] and 37.8 mm compared to Tejas et al. [35]. It is noteworthy that 52,.8% of the days analyzed were of aridity. The precipitation was higher than 2.10 mm/h (Figure 17d) in only 2% of days. As in other seasons of the year analyzed in this study,

it evidences that the scarce rainfall events that occurred during the dry period tend to be concentrated by the day and hour. Therefore, these episodes nourish the relationship with the entry of mTa in days preceded by the action of mTc, which intensely heat the atmosphere, causing the humidity of the Atlantic air mass, which manages to have the strength to travel to the south of the Amazon, to be poured quickly over Porto Velho.

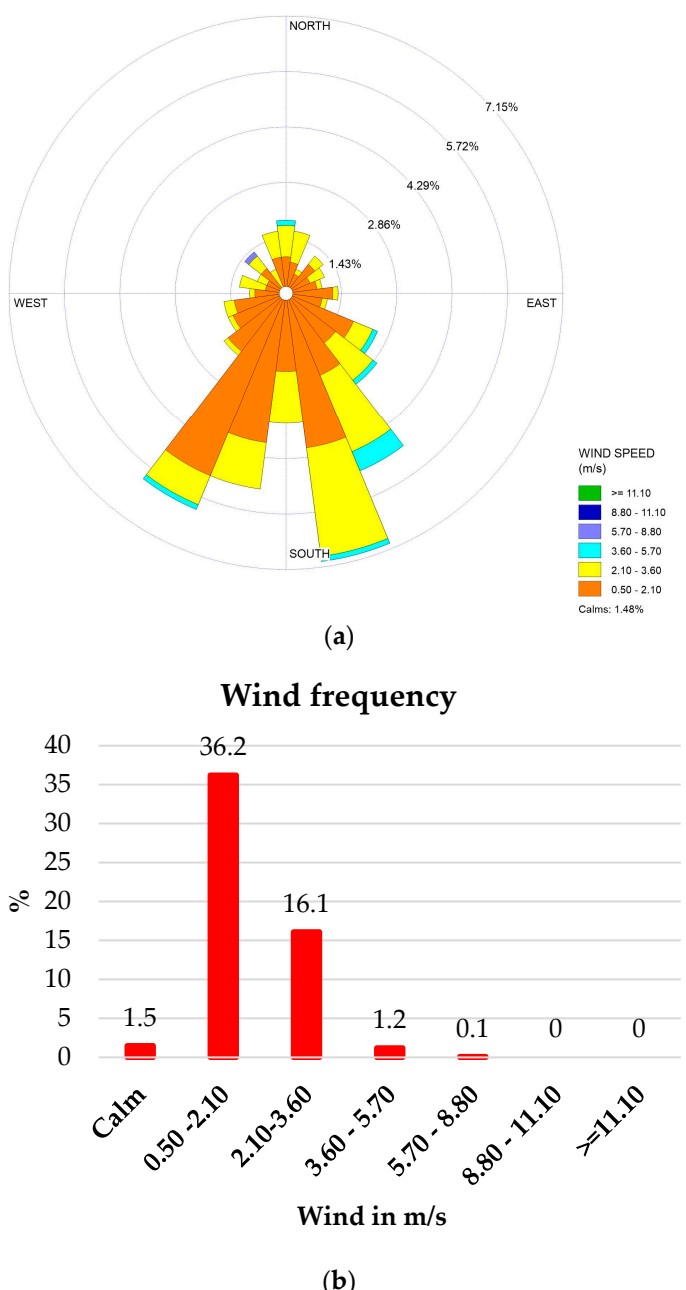

(**a**)

(**b**)

**Figure 17.** *Cont.*

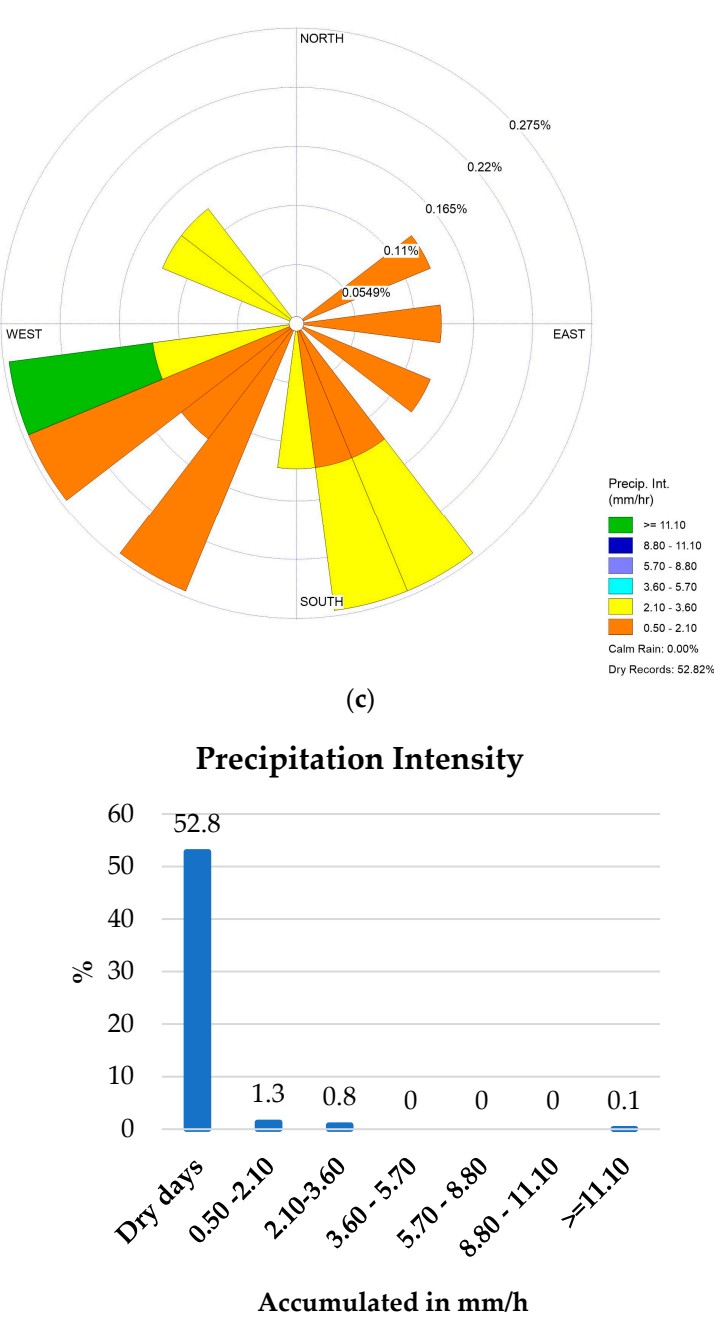

(**c**)

(**d**)

**Figure 17.** Wind rise (**a**) and frequency (**b**), rain rise (**c**) and intensity (**d**) for August 2018 in Porto Velho, Rondonia. Source: organized by the authors, based on Tejas [24], INMET [37], and Lakes Environment [36].

Figure 18a presents the correlation between the climatological variables in the period of higher drought, and the graph demonstrates the strong negative correlation between pressure and average temperature (r = −0.71) at a significance level of 0.001. This behavior is due to the atmospheric pressure influence on air temperature due to the presence of anticyclones in the study area.

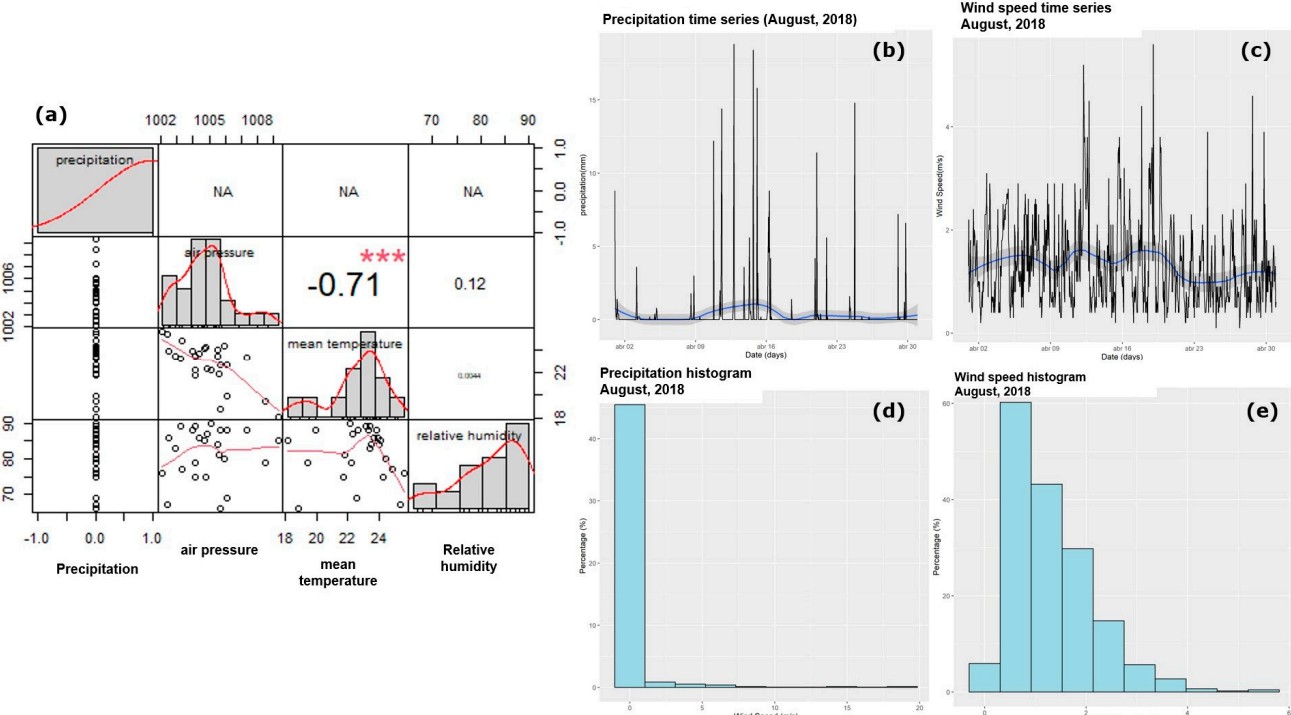

**Figure 18.** Brief exploratory analysis using the graphs (**a**) correlogram, (**b**) precipitation/day/month, (**c**) precipitation and wind speed/day/month, (**d**) precipitation/wind speed histogram, and (**e**) wind speed histogram for August 2018. Legend: *** Significant correlation at the 0.001 level. Source: organized by the authors.

Figure 18b,c represents the behavior and frequency (d,e) of the rain and wind variables for the period of August 2018, which is considered high air temperature and low humidity. With the Chi-Square test applied to the variables of air masses and rainfall, it was observed that the results presented a *p*-value of 0.10, with three degrees of freedom and an $X^2$ of 6.31. The significance level adopted for these results was 5%, so the null hypothesis ($H_0$) is not rejected, and a statistically significant association is not concluded between the variables (mEc, mTa) and the presence of rain, as shown in Figure 18b.

However, the start of the dry season triggers an increase in the hygrothermal amplitude, which would mean an increase in the negative correlation between the two climatic variables. However, the entry of the mPa favors the formation of a pre-frontal system, where temperatures tend to rise only to fall under the action of the mPa, while humidity during this process tends to remain stable with negative variations since, in many of these episodes, the mPa reaches the Western Amazon having discharged rainfall along its route across the South American continent. Therefore, the fact that the correlation between humidity and temperature did not remain negative can be explained by the rapid negative variation in temperature and humidity, as can be seen in Figure 16. Between 9 August and 10 August 2018, the average temperature fell by 1.4 °C, and humidity also fell by 15%.

When analyzing the relationship between wind direction and speed with precipitation data, it was concluded that the null hypothesis ($H_0$) should not be rejected since the data obtained by the test revealed a *p*-value (0.0004) with five degrees of freedom and $X^2$ (22.46). Also, Figure 18 shows precipitation frequency (>45%) with low wind speed (d), as well as the wind speed distribution for October, with (>40%) between 0 and 2.0 m/s (e).

The applied Chi-Square test revealed a *p*-value of 0.35, with 15 degrees of freedom and an $X^2$ of 16.54, to amplify the associations between wind direction and air masses. The occurrence of action of air masses (mEc, mTa) does not have a statistically significant dependence between the variables.

## 4. Discussion

Weather conditions in the Southwestern Amazon, at Porto Velho, Rondonia, in the transition season of October 2017, showed a low precipitation rate due to the predominance of the mTc air mass. Furthermore, Wright et al. [42] clarify that wet season onset in the tropics is generally associated with either monsoon reversals in the land–ocean temperature gradient or north–south migration of the Intertropical Convergence Zone (ITCZ), both of which are driven by seasonal changes in the distribution of solar radiation.

During the austral summer, the Southwestern Amazon is in the period with the highest rainfall, and this is due to the action of the mEc air mass (continental equatorial) added to the formation of the SACZ and the Bolivian High. This period was marked by rains more concentrated in some hours of the day and withal voluminous. Viana et al. [39] clarify that this event is characterized by continuous rain with a typical duration of 4 days that impacts regions during its action. According to Paccini et al. [2], the formation of the SACZ brings northerly moist winds from equatorial regions, especially the Atlantic Ocean. Another essential feature is an elongated northwest/southeast band of convection, particularly on the 4th, under the direct influence of the formation of a band of cloudiness caused by the SACZ. The record of rain in a single day had consequences in the urban environment of Porto Velho, Southwestern Amazon, for the formation of flooding and overflow of the streams that cut through the city. Espinoza et al. [43] point out that extreme rainfall events in the Amazon can be attributed to variations in sea surface temperatures (SST) in the tropical Pacific and Atlantic oceans. The study by Segura et al. [6] shows that Bolivian High evidenced a strong and significant relationship between December–March precipitation.

In the transition season from the rainy to the dry season, the weather conditions are influenced by the mEc type air mass and with some episodes of the entrance of the third branch of the air mass mPa, known regionally as coldness (*"friagem"*). Borsato and Mendonça [19] reinforce that mPa can advance to the Equator, causing this phenomenon in the Amazon, and the participation of mPa decreases towards the north in time of action and intensity. According to the statistics presented in the climate rhythm graph, April 2018 resulted in high temperatures, allied to the action of the air mass mTc.

Weather conditions in the Southwestern Amazon for August 2018 are of high air temperatures with episodes of cold due to the entry of the air mass mPa, in which the minimum result is 17 °C. Camarinha Neto et al. [44] point out two to three coldness per year, predominantly in the less rainy season in the central Amazon.

Another striking feature of this period is the reduction in rainfall, as the statistics presented in this article. The water scarcity could be potentiated by the deforestation process that the Southwestern Amazon has been going through. In an article about precipitation in Rondonia, Silva Dias et al. [45] explain that most of the results focused on the dry season effects of deforestation. Butt, Oliveira, and Costa [29] explain that the replacement of the forest by agricultural activities that do not maintain the humidity over larger areas could have effects powerful enough to disrupt rainfall patterns on a continental scale.

Anthropogenic activities modify natural climatic conditions, just as deforestation produced a peculiar climate for the Southwestern Amazon. The present paper analyzed the atmospheric patterns in the seasonality of October 2017 and August 2018, which instigates more detailed investigations of the impact of human activities, such as the removal of the rainforest and unbalancing the regional climate, especially the rainfall regime. Zemp et al. [46] point out that, as a result of deforestation, the factional reduction in evapotranspiration is stronger during the dry season (June–August) in the southern part of the Amazon Forest. And according to Mu, Biggs, and Sales [47], forests mitigate drought in the agricultural region, providing an important ecosystem service that further deforestation could disrupt.

That said, the analysis of climatic elements through the understanding of rhythm agrees with Armani and Galvani [18] on the geographical expression of climate and the causal relationships between atmospheric circulation and the temporal variation of at-

tributes, mainly because it allows the geographer to understand the situation of a small space in a global/zonal dimension.

## 5. Conclusions

The paper sought to analyze the weather conditions and the influence of air masses in the Western Amazon region in Porto Velho, Rondonia, through climatic rhythm. The methodological proposal enables the verification of the region's climatology during the seasonality from October 2017 to August 2018.

The Tropical Continental air mass (mTc) prevailed during the dry to rainy transition seasons (October). While in the rainy season (January), the action of the Continental Equatorial Mass (mEc) together with the South Atlantic Convergence Zone (ZCAS) predominated. In the transition between the rainy and dry seasons (April), the air mass (mEc) had reduced participation over the region, followed by the action of the Atlantic Tropical (mTa) and Continental Tropical (mTc) air masses. In August 2018, considered a dry period, the highest percentage of mTa took place, with episodes of the Atlantic Polar air mass (mPa) that contributed to the coldness phenomenon in the region.

Thus, the rainy periods are controlled by the mEc, and the drought is associated with the mTa. However, the mEc had a fundamental role in preventing the drought conditions from becoming more severe. It is worth mentioning that changes in use and occupation in the region, such as deforestation and fires, can potentiate the dry season, unbalancing the environmental services that the forest provides.

Therefore, this paper should serve as a basis for future research on rhythmic analysis in the Amazon region associated with the impacts of human activities such as tropical forest removal and its consequences for the regional climate.

**Author Contributions:** All authors contributed to the study's conception and design. Material preparation, data collection, and analysis, G.T.T., R.M.S.S., and M.R.F.; first draft, G.T.T.; All authors commented on previous versions of the manuscript. All authors have read and agreed to the published version of the manuscript.

**Funding:** This research was funded by the Coordination for the Improvement of Higher Education Personnel Foundation (CAPES—00.889.834/0001-08) in the granting of a PhD scholarship of the Regional Development and Environment Program (PGDRA) of the Federal University of Rondonia (UNIR).

**Data Availability Statement:** The datasets generated during and/or analysed during the current study are available in the National Institute of Meteorology (INMET) repository, https://portal.inmet.gov.br/dadoshistoricos, accessed on 12 December 2018; Weather Prevision Center and Climate Studies (CPTEC/INPE) http://tempo.cptec.inpe.br/boletimtecnico/pt (acessed on 20 November 2017); Brazilian Navy—Navy Hydrography Center—CHM, Synoptic letters, http://marinha.mil.br/chm/dados-do-smm-cartas-sinoticas/cartas-sinoticas, accessed on 16 October 2018; Access to RitmoAnalysis software, version 2.0 (Borsato, V. A. A Participação dos sistemas atmosféricos atuantes na bacia do rio Paraná no período de 1980 a 2003. Tese (parcial), (Doutorado) Nupélia, Universidade Estadual de Maringá. Maringá, 2006. Borsato, V. A. Borsato F. H. A dinâmica atmosférica e a influência da tropicalidade no inverno de 2007 em Maringá PR—Espacial. In 8° Simpósio Brasileiro de Climatologia Geográfica. Evolução Tecnológica e Climatologica, Universidade Federal de Uberlândia, Agosto 2008. Eixo 5—Técnica em Climatologia—CD-ROM); Access to WRPLOT View, https://www.weblakes.com/software/freeware/wrplot-view (accessed on 1 August 2020), version 8.0.2.

**Acknowledgments:** The authors would like to thank CAPES for encouraging the development of this research through a doctoral scholarship. The authors Galvani, E.; Gobo, J.P.A. would also like to thank CNPq for supporting the development of the research and for the Research and Productivity grant.

**Conflicts of Interest:** The authors declare no conflicts of interest.

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
