# Peer review of "Atmospheric Patterns in Porto Velho, Rondônia, Southwestern Amazon, in a Rhythmic Context between 2017 and 2018"

_climate, doi:10.3390/cli12030028_

Round 1

Reviewer 1 Report

Comments and Suggestions for Authors

I have reviewed the paper, and I have 3 major comments which I think merit major revisions:

1. After reading this paper and one of the other English language references it is still very unclear to me what is mean by rhythm analysis. It looks like what the authors are doing is something along the lines of a manual synoptic air mass classification, which they then use to understand day to day weather variability. There's nothing wrong with this, but it would aid many readers if the technique was compared to other techniques, or a more comprehensive description of it was included.

2. More clarity is needed on the air mass classification step. If there are not quantitative guidelines for establishing the different air masses and synoptic systems including some images of  representative systems would be very useful in understanding the different air mass characteristics.

3. In general the quality of the figures is very low. Much of the text is unreadable at the size it is currently rendered at, and many of the figures have not been rendered in sufficiently high resolution to make the details clear.

4. The analysis references the different significant tests and correlations between pressure, humidity, temperature, and precipitation. Some explanation of how these correlations relate to the different air masses and meteorological processes is needed in order to increase the significance of the paper.

Author Response

First of all, we would like to thank you for your comments on this manuscript. We will respond to each of your comments below:

  1. Thank you for drawing attention to the term rhythm analysis. It is a methodological procedure to understand the daily sequence of atmospheric phenomena in a given place and to establish an analysis based on homogeneous data, which represents the proximity of a "usual" rhythm or an exceptionality, which would be the irregularities. The manuscript highlights what rhythm is.

  1. That the air masses were classified using synoptic charts and GOES-16 satellite images To record the atmospheric systems, tables were drawn up in an Excel spreadsheet. Numerical values (24) were assigned to the day when a single system acted in the region, and sometimes (12) for each, when the region was under the confluence of two systems, or different values, judging by the time of participation. The monthly or seasonal values were considered in percentages, and these in turn in histograms and cartograms.

  1. The figures have been corrected for greater visualization.

  1. The aim was to evaluate the number of days on which each air mass caused an increase or decrease in climatological variables. There are various applications for the Chi-squared test. In this case, it was used to check whether two variables are dependent (alternative hypothesis) or independent (null hypothesis). The result of the test alone does not attest to cause and effect. We can only state that there is a relationship between two variables. By default, statistical tests are designed to rule out the null hypothesis. The idea is: if I reject the null hypothesis, what is the chance that I'm wrong? p-value represents the chance of making that mistake. Usually a risk of less than 5% is assumed (p-value <0.05). In these cases, if the test gives a p-value of >0.05, we can't rule out the null hypothesis. Below, I have manipulated, in the honest sense of the word, the data in such a way that the answers are binary: yes or no. This way, the results are clear. This way, the results are clear. Example: First test below. Does the presence of rain depend on the presence of air mass X or Y? Count the number of Yes or No answers and run the test. If the p-value < 0.05, we reject the null hypothesis. In other words, the incidence of rain depends on the air mass.

Reviewer 2 Report

Comments and Suggestions for Authors

The authors took efforts to collect the surface climate, air mass data, chart, etc., to demonstrate the character of weather conditions. 

But I'm not sure what kind of new and scientific knowledge can we get from the analysis, or how the analysis benefits the the communities outside the climatologists and meteorologists.  In addtion, the subject of submission does not fit the scope of journal, as it used only one-season-cycle data from one weather station, which is too coarse for understanding the weather episode or phenomenon, and certainly not containning enough data and analysis to provide regional climatic information. 

Obviously the shows that High Bolivia has evidenced a strong and significant relationship between 526 December–March Precipitation. 527

In the transition season from the rainy to the dry season, the weather conditions are 528 being influenced by the mEc type air mass and with some episodes of the entrance of the 529 3rd branch of the air mass mPa, known regionally as coldness (“friagem”). Borsato and 530 Mendonça [19] reinforce that mPa can advance to the Equator, causing this phenomenon 531 in the Amazon, and the participation of mPa decreases towards the north in time of ac-532 tion and intensity. According to the statistics presented in the climate rhythm graph, 533 April 2018 resulted in high temperatures, allied to the action of the air mass mTc. 534

Weather conditions in the Southwestern Amazon for August 2018 are of high air 535 temperatures with episodes of cold due to the entry of the air mass mPa, in which the 536 minimum result is 17°C. Camarinha Neto et al. [44] point out two to three coldness per 537 year, predominantly in the less rainy season in the central Amazon. 538

Another striking feature of this period is the reduction in rainfall, as the statistics 539 presented in this article. The water scarcity could be potentiated by the deforestation 540 process that the southwestern Amazon has been going through. In an article about pre-541 cipitation in Rondonia, Silva Dias et al. [45] explain that most of the results have focused 542 on the dry season effects of deforestation. Butt, Oliveira, and Costa [29] explain that the 543 replacement of the forest by agricultural activities that do not maintain the humidity over 544 larger areas could have effects powerful enough to disrupt rainfall patterns on a conti-545 nental scale. 546

Anthropogenic activities modify natural climatic conditions, just as deforestation 547 has produced a peculiar climate for the Southwestern Amazon. The present paper ana-548 lyzed the atmospheric patterns in the seasonality of October 2017 and August 2018, 549 which instigates more detailed investigations of the impact of human activities, such as 550 the removal of the rainforest and unbalancing the regional climate, especially the rainfall 551 regime. Zemp et al. [46] point out that, as a result of deforestation, the factional reduction 552 in evapotranspiration is stronger during the dry season (June–August) in the southern 553 part of the Amazon Forest. And according to Mu, Biggs, and Sales [47], forests mitigate 554 drought in the agricultural region, providing an important ecosystem service that further 555 deforestation could disrupt. 556

The local weather pattern is obviously influenced by climate patterns. In the paper the air mass was an important elment to explain the weather condition. And even though the data was download from internet (the link in line 131 does not work however), I suppose the authors should provide the methoological details and conduct evaluation before connecting it with local weather conditions.  And has the auto station data been through quality control? 

The description of figs 2,6,10 are not complete. 

Comments on the Quality of English Language

The writing is generally good. But the numbers in the form such as "2,10 mm/h" are not regular in scientific paper.  

Author Response

First of all, we would like to thank you for your comments on this manuscript. We will respond to each of your comments below:

With regard to the use of just one weather station, we would like to point out that the INMET station (A-925), WMO code 81932, located on the premises of the Brazilian Agricultural Research Corporation (EMBRAPA) (8°47'S, 63°50'W, 87m) is the only one with reliable historical data for the city of Porto Velho, Rondônia. This is a problem for the Amazon region, i.e. the lack of reliable data to support climatological research. As the aim of this article was to understand the action of atmospheric systems in the city of Porto Velho and considering that the spatial scope of the weather station is 30 km, we believe that there is no distortion or methodological breach in using data from a single station. Due to the lack of articles on the study area, this essay sheds light on the behaviour of previously unorganized and even unknown atmospheric systems, which could help inform other studies.

We have made the necessary adjustments to Figures 2, 6 and 10.

With regard to separating decimal places with a comma "," we used the standard of the international system of units. In this case, we separated the decimal places with a comma "," and the thousand with a period "."

Reviewer 3 Report

Comments and Suggestions for Authors

1. WEAKNESES

There is a typical case for mainstream paper in the climatology field. It helps to became  financial grants, but not helps by my opinion enough for the  serious extending of knowledge in the thematic of the study  (in this form, which is presented there). The accent in the “Discussion” section is on the anthropogenic effects over regional climate in south-west periphery of  Amazonia (mainly deforestation).  However the base for such one needs of data analysis at least for  20-30 years or even much better for longer intervals- for example since t’ middle of 20th century , before the modern economic development of the region.  Thus a comparison  of type “before” and “after”  the deforestation process could be possible. The correct study should includes preliminary extracting from the climate parameters (temperature, precipitations, atmospheric pressure etc) time series  the trends and cycles caused by natural factors changes, such as solar and volcanic activity, ENSO phenomena etc. After the removing of the above mentioned influences , the analysis of  eventual forcing by non-natural factors is possible. It is obviously that  an analysis , which is based on data only of single annual season period (2017-2018) is absolutely not enough for analysis and discussion about  any climate factor influence, including also the possible anthropogenic effects- the period 2017-2018 describes only a moment picture of regional climate and no any changes related to the last one!  But is it possible to expect the same climate situation in Portu Velho for example 3 years later in 2020 – during LaNinja phenomena and 11yr solar minimum?! … Most probably  –not!  That’s why any discussions for the possible anthropogenic or other influences over Rondonia climate are inadequate there , the real subject of this work are the climate conditions in Portu Velcho during the  concrete annual cycle.

2. THE POSITIVE ASPECTS

The used statistical methods are adequate for the aims of this study. The obtained results seem  realistic. The English is too simple and clear (for me).  No problems for understanding of the text.

3.RECOMMENDATION 

It is clear that the manuscript needs of very grand revision. However it is clear for me, that this one will be not made. The rejecting is also  not good decision, but I will not explain why?! .  I could  recommend in this case:

  1. The Section “Discussion” needs of reworking. Especially comments about the natural factor influences (see 1) should  to be added.
  2. The  font size of text  (legends)  in the figures is too small. It is difficult to read it on many places. Please, enlarge the font size in the figures.

Author Response

First of all, we would like to thank you for your comments on this manuscript. We will respond to each of your comments below:

In fact, the focus of the article submitted was to understand the action of atmospheric systems in Porto Velho, Rondônia, during an annual cycle. When we drew attention to the low rainfall in August, the dry season, we used other studies to demonstrate that this is indeed the critical month for rainfall. The studies we cited not only demonstrate this, but also draw attention to the decrease in rainfall in August. Our aim is not to demonstrate or explain whether deforestation is influencing rainfall rates. However, our work clearly demonstrates that there was a relationship between the action of certain atmospheric systems and the decrease in the amount of rain in the period analyzed.

With regard to the influence of natural phenomena, we have tried to make changes so that the results reflect the behavior of climate variables as a function of the action of atmospheric systems.

The figures have been corrected for greater visualization.

Round 2

Reviewer 1 Report

Comments and Suggestions for Authors

1. I think that it would be helpful for English language readers to have a 1-2 paragraph description of rhythm analysis, and how it is applied specifically to this study.

2. A table should be included (possibly in supplementary materials) giving specific criteria for the classification of the different air masses. If no such criteria exists (which is difficult to discern from the authors' comments), then some representative examples (e.g. weather maps or satellite images on a day when a particular air mass is present) should be included so that the readers can understand the classification system and what the properties of the air masses are.

3. I note that wind roses, rain roses, and the correlegrams with associated time series plots have still mostly illegible text. The figures showing precipitation time series have the wrong axes labels.

4. To clarify, I am asking the authors to give a physical explanation as to the statistically significant relationships that they have found differ between seasons.  For instance, in January and April 2018  mean temperature and relative humidity have statistically significant negative correlation, whereas in August 2018 they do not. Explaining how this relates to the meteorological conditions present during the different seasons would greatly strengthen the manuscript.

Author Response

First of all, we would like to thank you for your comments on this second round of review.
1. An explanation of rhythmic analysis has been added as suggested;
2. Satellite image, synoptic chart and diagram of the vertical profile of the atmosphere have been added;
3. The figures have been improved as suggested;
4. As suggested, an explanation has been added for the difference between the correlations between humidity and temperature for the transitional and dry seasons.  

Round 3

Reviewer 1 Report

Comments and Suggestions for Authors

Thank you for addressing the comments, the figures are especially improved.

Author Response

We welcome your comments to improve the paper.